# CdbA is a DNA-binding protein and c-di-GMP receptor important for nucleoid organization and segregation in *Myxococcus xanthus*

Dorota Skotnicka[1], Wieland Steinchen[2,3], Dobromir Szadkowski[1], Ian T. Cadby [4], Andrew L. Lovering [4], Gert Bange[2,3] & Lotte Søgaard-Andersen [1✉]

Cyclic di-GMP (c-di-GMP) is a second messenger that modulates multiple responses to environmental and cellular signals in bacteria. Here we identify CdbA, a DNA-binding protein of the ribbon-helix-helix family that binds c-di-GMP in *Myxococcus xanthus*. CdbA is essential for viability, and its depletion causes defects in chromosome organization and segregation leading to a block in cell division. The protein binds to the *M. xanthus* genome at multiple sites, with moderate sequence specificity; however, its depletion causes only modest changes in transcription. The interactions of CdbA with c-di-GMP and DNA appear to be mutually exclusive and residue substitutions in CdbA regions important for c-di-GMP binding abolish binding to both c-di-GMP and DNA, rendering these protein variants non-functional in vivo. We propose that CdbA acts as a nucleoid-associated protein that contributes to chromosome organization and is modulated by c-di-GMP, thus revealing a link between c-di-GMP signaling and chromosome biology.

[1] Department of Ecophysiology, Max Planck Institute for Terrestrial Microbiology, Karl-von-Frisch Str. 10, 35043 Marburg, Germany. [2] Center for Synthetic Microbiology, Hans-Meerwein Str. 6, 35043 Marburg, Germany. [3] Faculty of Chemistry, Philipps-Universität Marburg, Hans-Meerwein Str. 4, 35032 Marburg, Germany. [4] School of Biosciences, University of Birmingham, Edgbaston, Birmingham B15 2TT, UK. ✉email: sogaard@mpi-marburg.mpg.de

In bacteria, nucleotide-based second messengers fulfill key functions in the generation of output responses to changing environmental and cellular cues[1]. Among them, cyclic dinucleotide bis-(3′-5′)-cyclic dimeric GMP (c-di-GMP) stands out as widespread and highly versatile regulating a multitude of diverse processes including biofilm formation, motility, adhesion, exopolysaccharide (EPS) synthesis, cell cycle progression, development, and virulence[2,3]. Signal transduction by c-di-GMP involves its regulated synthesis and degradation by diguanylate cyclases (DGCs) and phosphodiesterases (PDEs), respectively while output responses are generated by c-di-GMP binding to and allosteric regulation of downstream receptors[2,3]. Receptors include riboswitches and proteins that regulate processes at the transcriptional, translational or post-translational level[3]. Proteinaceous receptors comprise a variety of proteins with no or little sequence homology including catalytically inactive DGCs and PDEs[4–8], PilZ-domain proteins[9–15], MshEN domain proteins[16,17], different transcription factor families[18–21] and ATPases of flagella and type III as well as type VI secretion systems[22]. Typically, bacterial genomes encode multiple DGCs and PDEs[3,23]. For some DGCs and PDEs distinct functions have been defined; however, for many, no function has been assigned. Similarly, the number of DGCs and PDEs often exceeds that of known c-di-GMP receptors impeding a complete understanding of the biological functions of c-di-GMP signaling and how effects of c-di-GMP are implemented. These observations also support that additional functions of c-di-GMP regulation remain to be uncovered.

The Gram-negative deltaproteobacterium *Myxococcus xanthus* is a model organism for studying social behavior in bacteria[24]. In the presence of nutrients, *M. xanthus* cells generate coordinately spreading colonies[24]. In response to nutrient depletion, development is initiated resulting in the formation of multicellular, spore-filled fruiting bodies[25]. c-di-GMP accumulates during growth and development and the level increases ~10-fold during development[26,27]. During growth, c-di-GMP regulates T4P-dependent motility and the composition of the extracellular matrix[6,26]. During development, the increase in c-di-GMP level is essential for EPS accumulation, fruiting body formation, and sporulation[27]. Little is known about c-di-GMP receptors in *M. xanthus* and only the NtrC-like transcriptional regulator EpsI/Nla24, which is important for motility and EPS accumulation[27,28], and the histidine protein kinase SgmT, which possesses an enzymatically inactive DGC domain and regulates extracellular matrix composition[6], have been identified as c-di-GMP receptors.

During growth, *M. xanthus* cells divide at midcell[29,30] and each daughter cell contains a single, fully replicated chromosome with the origin and terminus regions in the subpolar regions close to the old and new cell pole, respectively[31–33]. Replication and chromosome segregation is initiated soon after cell division[31]. Segregation depends on the ParABS system in which ParB binds to *parS* sites close to the origin while the ParA ATPase and ParB mediate segregation[31,32]. After duplication of the origin region, one ParB/*parS* complex remains in the subpolar region of the old pole while the second copy translocates to the subpolar region of the new pole. In parallel, the terminus region translocates to midcell where it is replicated[31]. In the subpolar regions, the ParB/*parS* complexes are anchored to a scaffold composed of the three BacNOP bactofilins and the PadC adaptor protein that together form a complex that extends ~1 μm away from the cell pole[33,34]. The ParABS system is essential for chromosome segregation[31]. By contrast, BacNOP and PadC anchor the origin region and in their absence the nucleoid is more compact but BacNOP and PadC are not essential[33].

Here, we identify two small, paralogous proteins, CdbA and CdbB, as c-di-GMP receptors. We show that CdbA is a tetrameric ribbon-helix-helix DNA binding protein and changes conformation upon c-di-GMP binding. The regions in CdbA important for c-di-GMP binding are also important for DNA binding. Consistently, c-di-GMP and DNA binding are mutually exclusive. In vivo CdbA is essential, while CdbB is not, and depletion of CdbA causes defects in chromosome organization and segregation resulting in defects in cell division. CdbA is abundant and binds globally to the *M. xanthus* chromosome. These observations are in agreement with a model whereby CdbA is a ligand-regulated nucleoid-associated protein (NAP) important for chromosome organization and segregation and in which CdbA activity is modulated by c-di-GMP.

## Results

**CdbA and CdbB bind c-di-GMP in vitro.** We identified c-di-GMP binding proteins using the unbiased c-di-GMP capture compound technology[35] and cell extracts of growing wild-type (WT) *M. xanthus* cells (Methods). Here, we focused on MXAN_4361 and MXAN_4362 (renamed CdbA and CdbB, respectively for c-di-GMP binding protein A and B), which were enriched in the experimental samples compared to the controls, due to their lack of homology to known c-di-GMP receptors and because in vivo analyses demonstrated that CdbA is essential (see below).

CdbA and CdbB are small paralogs (67 and 86 amino acids, respectively; 50.6/57.3% identity/similarity) and encoded in an operon (Fig. 1a, Supplementary Fig. 1a–c). All fully sequenced Myxococcales genomes encode at least one ortholog, and when only one is present, then based on sequence identity/similarity, it is CdbA-like (Supplementary Fig. 1b). CdbA/CdbB homologs were not identified outside the Myxococcales. We confirmed the interaction with c-di-GMP employing purified CdbA and CdbB proteins in a differential radial capillary action of ligand assay (DRaCALA)[36], where both C-terminally His$_6$-tagged CdbA and CdbB specifically bound $^{32}$P-labeled c-di-GMP (Fig. 1b), i.e., binding was outcompeted by excess unlabeled c-di-GMP, but not by unlabeled GTP (Fig. 1b). Based on isothermal titration calorimetry (ITC), CdbA binds c-di-GMP with high affinity ($K_d$ of ~83 nM) and a stoichiometry of 0.5 molecules of c-di-GMP per 1 molecule of CdbA (Fig. 1c). By analytical size-exclusion chromatography (SEC) in the absence of c-di-GMP, CdbA had an apparent molecular mass of ~38 kDa suggesting that CdbA is a tetramer (Fig. 1d). In the presence of c-di-GMP, CdbA had an apparent molecular mass of ~52 kDa (Fig. 1d) suggesting that c-di-GMP binding does not alter CdbA oligomerization but modulates the conformation of tetrameric CdbA resulting in a more open conformation than in the unbound form.

**CdbA has a ribbon-helix-helix fold.** CdbA and CdbB do not share sequence homology with known c-di-GMP binding proteins; however, sequence analysis suggested that they belong to the ubiquitous ribbon-helix-helix (RHH) superfamily of transcription factors, named so after the order of secondary structure elements (β-strand/α-helix1/α-helix2) in each subunit of the dimeric RHH DNA binding domain[37]. In this domain, two intertwined chains form a β-sheet consisting of the two antiparallel β-strands that insert into the DNA major groove to make sequence-specific contacts. Additionally, the loop between α-helix1 and α-helix2 together with the N-terminus of α-helix2 make contacts to the DNA phosphate backbone[37]. RHH proteins typically bind DNA as tetramers recognizing direct or inverted repeat sequences[37].

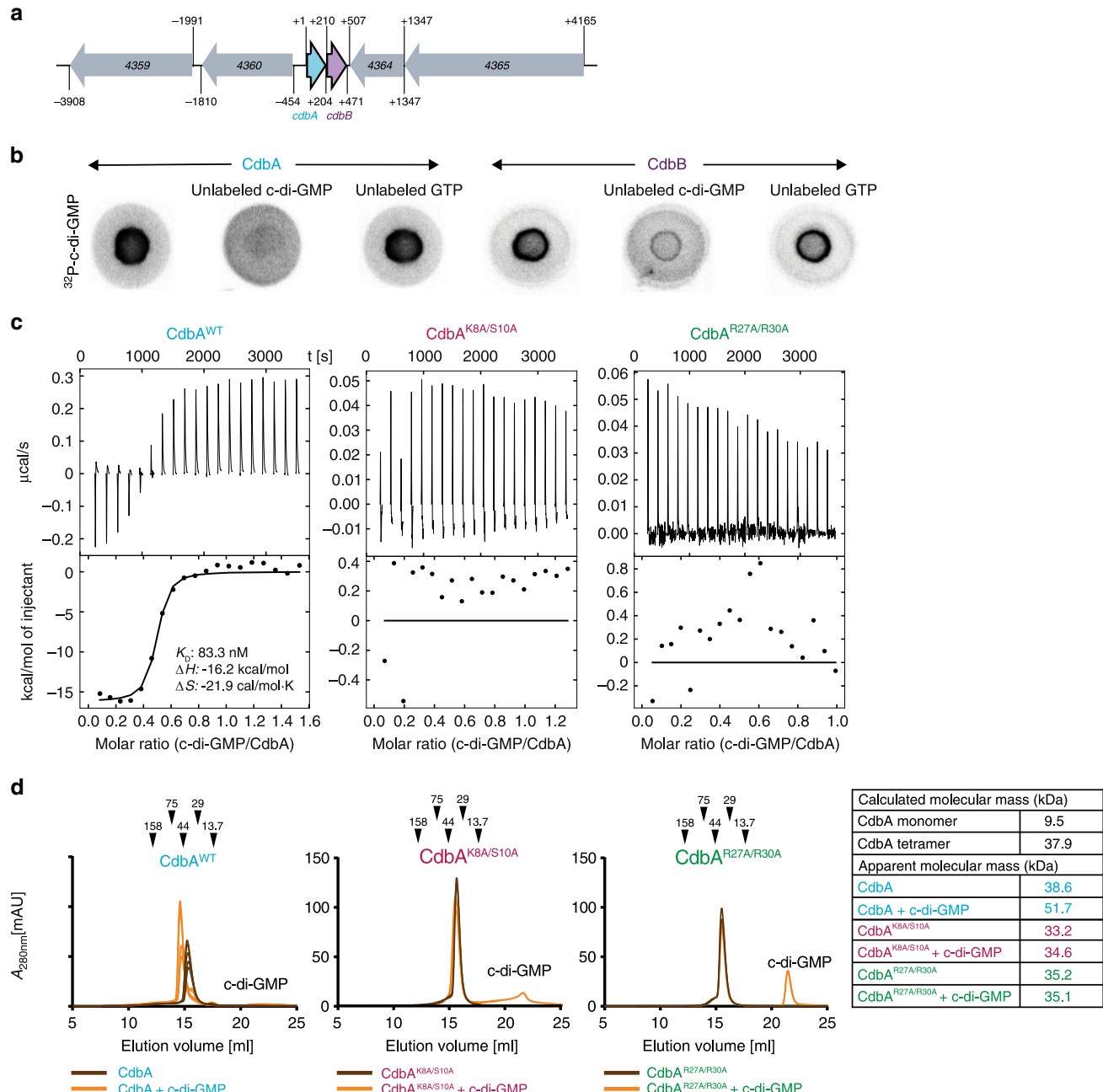

**Fig. 1 CdbA and CdbB bind c-di-GMP in vitro. a** *cdbA-B* locus. Start and stop codons are indicated. +1 indicates first nucleotide in the *cdbA* start codon. Arrows indicate direction of transcription. MXAN locus tags are indicated for the genes flanking *cdbA-B*. See Supplementary Fig. 1 for annotation of these genes. **b** CdbA and CdbB binding to $^{32}$P-c-di-GMP in DRaCALA in the absence and presence of unlabeled c-di-GMP and GTP. Similar results were obtained in two independent experiments. Source data are provided as a Source Data file. **c** ITC of c-di-GMP-binding to CdbA variants. In each experiment, the original titration traces (top panel) and integrated data (bottom panel) are shown. Solid line in the bottom panel represents the fit of the integrated titration peaks to a one-site binding model. Dissociation constants ($K_D$) and thermodynamic parameters are listed for CdbA$^{WT}$ for which binding was detected. **d** Analytical SEC of CdbA variants in the presence and absence of c-di-GMP. Arrows indicate elution volume and mass of molecular weight standards. $n = 5$ for CdbA$^{WT}$, $n = 2$ for CdbA$^{K8A/S10A}$ and CdbA$^{R27A/R30A}$. Table indicates calculated and apparent molecular mass of CdbA variants. Source data are provided as a Source Data file.

To gain insights into the mechanism of CdbA and CdbB, we determined the CdbA crystal structure (Supplementary Table 1). We obtained two crystal forms of full-length CdbA-His$_6$. The first structure (traced residues 5–52) verified that CdbA is a RHH superfamily member, with crystallographic symmetry generating a classical RHH dimer (Fig. 2a). This observation was confirmed by DALI analysis[38], which suggested the streptococcal RHH repressor CopG as the closest structural homolog (PDB: 2CPG, r.m.s.d. of 1.9 Å over 45 residues[39]). In CdbA, the exposed face of the β-strand projects residues K8, S10 and Y12, while L11 and F13 are facing the hydrophobic core of the dimer. The β-strand is terminated by P14, leading into α-helix1 (residues 16–28), followed by a loop and then α-helix2 (residues 32–48), with a strongly kinked region at A43/R44, and a C-terminal tail traceable to A52 (Fig. 2a). The two α2 helices run antiparallel, with kink residues I42 and A43 packing against their counterparts in the opposing chain (Fig. 2a).

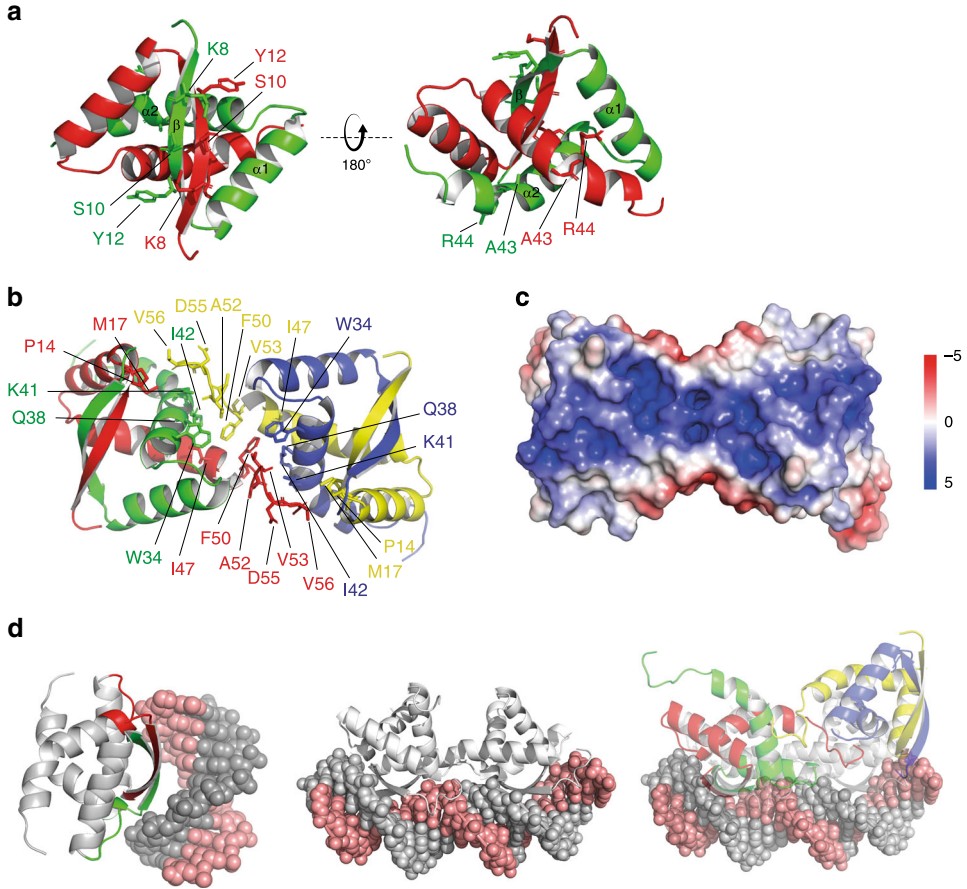

**Fig. 2 CdbA has a RHH fold. a** CdbA dimer with monomers in red and green. Residues discussed in text are indicated. **b** CdbA tetramer with individual monomers in red, green, yellow, and blue. Residues important for tetramer formation are indicated. **c** Electrostatic surface potential of CdbA. The electrostatic potential ranges from −5 (red) to +5 (blue) kT/e. The view of CdbA is as in **b**. **d** Model for DNA binding by CdbA. Left panel, CdbA dimer interacting with DNA with the two interfaces identified by HDX to be involved in c-di-GMP binding colored in red and green (see also Fig. 3e); middle panel, Arc tetramer bound to DNA (1BDT[41]); right panel, overlay of Arc tetramer bound to DNA and CdbA tetramer.

Successful model building of the first form allowed us to solve the second form via molecular replacement. This form has three copies of the protein in the asymmetric unit, one of which forms a tetramer via crystallographic symmetry (four copies of same chain) and two of which form a dimer within the asymmetric unit, which then tetramerizes over the final crystallographic axis. These two tetramers are independent yet essentially identical and we interpret these to represent the physiological tetramer identified by SEC. The tetramer centers on residue F50 of the two innermost chains, and the C-terminal tails can be traced five residues further than in form one, and with the C-terminal tail of the innermost chains in a dimer contacting an opposing dimer (Fig. 2b). Protein–protein contacts mediated by the tail region include a hydrophobic component from F50 and V53 in one chain projecting towards a pocket in the opposing dimer formed by W34/I42/I47, a second hydrophobic interaction between V56 and P14/M17, and a more polar interface between the free carbonyl groups of A52/V53/D55 and the side chains of Q38 and K41 (Fig. 2b). Upon solving this crystal form, we realized that a similar, but more loosely-packed, tetramer can be observed when considering the crystal symmetry of form one. The arrangement of the two dimers in the tetramer places their β-sheets ~35 Å apart (measured between Cα atoms of equivalent Y12 residues).

Using the RHH protein Arc, which binds DNA as a tetramer and with the two β-sheets separated by 26 Å[40,41], as a template, we created a model of how CdbA could bind DNA (Fig. 2c, d).

Consistent with the idea that the two β-sheets in the tetramer make DNA contacts, a surface electrostatic map of CdbA revealed that this region is positively charged (Fig. 2c). Because the β-sheets are ~35 Å apart in CdbA, this model suggests that CdbA bends DNA upon binding (Fig. 2d).

**c-di-GMP binding to CdbA induces conformational changes.** Because we were unsuccessful in obtaining the crystal form of c-di-GMP-bound CdbA, we performed hydrogen-deuterium exchange (HDX) mass-spectrometry (MS) in the presence and absence of c-di-GMP to identify regions in CdbA involved in c-di-GMP binding or undergoing conformational changes upon c-di-GMP binding. We obtained a total of 68 peptides in both states that cover the entire CdbA sequence with ~11-fold redundancy per amino acid (Supplementary Data 1, Supplementary Fig. 2a). The HDX profile in the absence of c-di-GMP is in agreement with the crystal structure, i.e., the RHH fold (amino acids 1–48) exchanges hydrogen to deuterium more slowly than the disordered C-terminal region (Supplementary Fig. 2a). The difference in HDX between c-di-GMP-bound and apo-CdbA revealed changes that were restricted to the RHH fold (Fig. 3a, Supplementary Fig. 2b). Specifically, the N-terminus, the β-strand, a portion of α-helix1, the loop between α-helices1 and 2, and the N-terminal tip of α-helix2 had reduced HDX in the presence of c-di-GMP (Fig. 3a–c). When illustrating the location of three

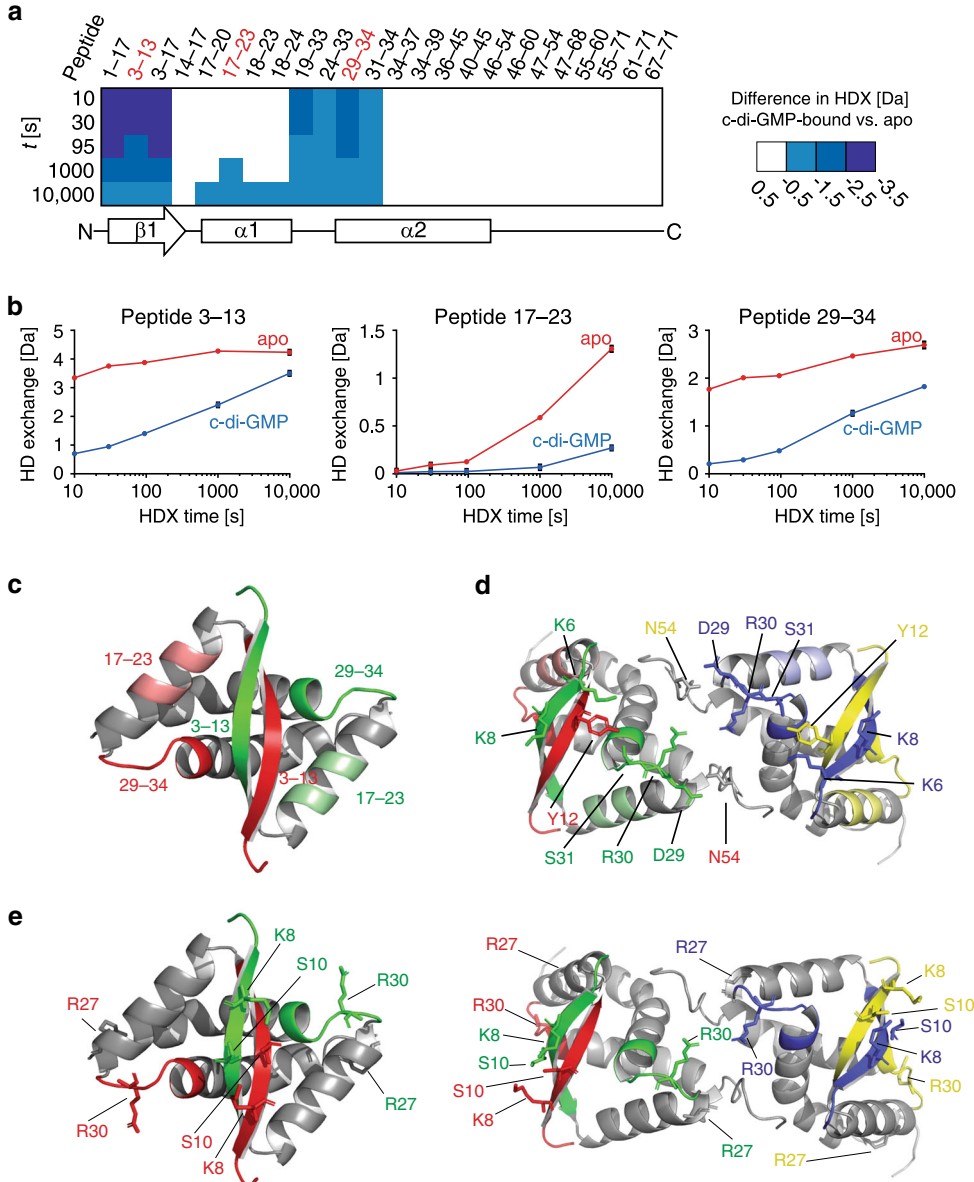

**Fig. 3 HDX assay for CdbA in the presence and absence of c-di-GMP. a** Representative peptides of CdbA are colored according to the difference in HDX between the c-di-GMP-bound and the apo-state. The secondary structure of CdbA is shown below. **b** HDX for three representative peptides of CdbA (marked red in panel **a**) in the c-di-GMP-bound (blue) and apo-state (red). Data represent mean ± SD ($n = 3$ separate reactions). **c** Location of the three representative peptides in **b** in the crystal structure of the CdbA dimer. **d** Location of the three representative peptides from **b** in the crystal structure of the CdbA tetramer with residues proposed to be involved in formation of the polar pocket for c-di-GMP binding indicated as sticks. **e** Crystal structure of CdbA dimer (left) and tetramer (right) with interface 1 and -2 of each monomer colored in red, green, yellow, and blue. Amino acids selected for site directed mutagenesis are indicated.

representative peptides that exhibit substantial differences in deuterium incorporation (Fig. 3b) onto the crystal structure of a CdbA dimer or tetramer (Fig. 3c), two of the three peptides (3–13 and 29–34) converge providing a polar pocket formed by residues Y12 and N54 from the two innermost chains and K6, K8, D29, R30, and S31 from the two outermost chains (Fig. 3d). Because HDX-MS cannot discriminate between peptides originating from the inward- and outward-facing chains, we cannot dissect whether the copies of the two peptides that do not engage in formation of this pocket have altered HDX profiles upon c-di-GMP binding. Nevertheless, given the proximity of the β-strand (peptide 3–13) and the helix α1–α2 loop including the N-terminal tip of α-helix2 (peptide 29–34), we hypothesized that they might represent the c-di-GMP binding site. From hereon, we

refer to these two regions as interface −1 and −2, respectively (Fig. 3e).

To probe the participation of interface −1 and −2 in c-di-GMP binding, we substituted residues in both interfaces that show 100% conservation in CdbA homologs (Supplementary Fig. 1c, Fig. 3e), to generate CdbA[K8A/S10A] and CdbA[R27A/R30A] (Fig. 3e). In SEC, both variants, eluted with an apparent molecular mass of ~35 kDa (Fig. 1d) indicating correct tetramer formation; however, they did not detectably bind c-di-GMP as estimated by ITC (Fig. 1c) and SEC (Fig. 1d) suggesting involvement of both interface-1 and -2 in c-di-GMP binding. Of note, the two interfaces important for c-di-GMP binding overlap with the regions of CdbA predicted to be involved in DNA binding (Fig. 2d).

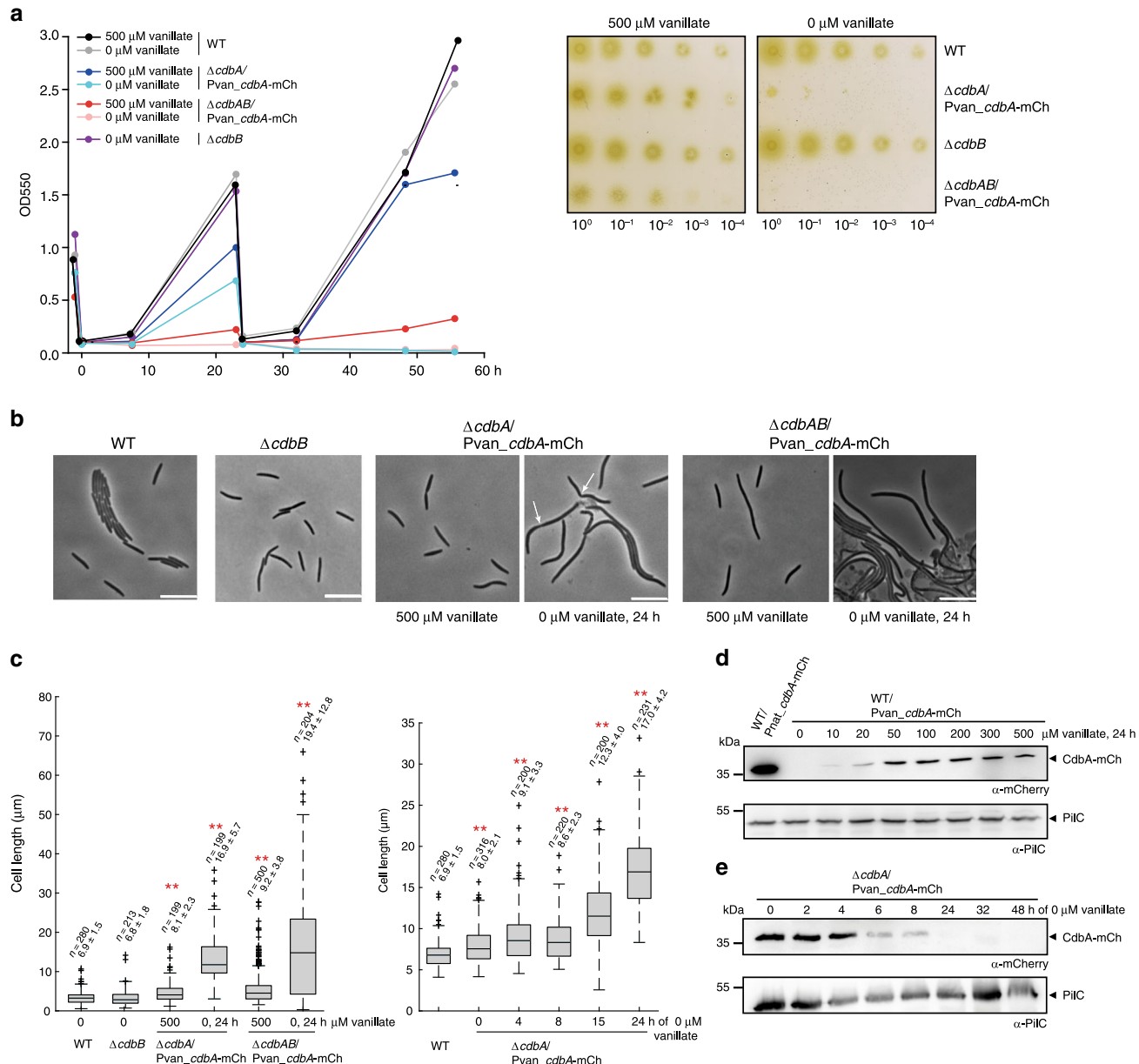

**Fig. 4 CdbA is essential for growth and CdbB is dispensable. a** Growth of indicated strains in suspension (left) and on solid surface (right) in the presence and absence of vanillate driving synthesis of CdbA-mCh. In the plating assay, plates were incubated for 4 days. Similar results were obtained in two independent experiments. Source data are provided as a Source Data file. **b** Phase contrast microscopy of representative cells of indicated genotypes grown in the presence or absence of vanillate. Cell were grown ±vanillate for 24 h. Arrows indicate constrictions. Scale bars, 10 μm. Similar results were obtained in two independent experiments. **c** Cell length distribution of cells of indicated genotypes grown in the presence or absence of vanillate. Left panel, cell length distribution of cells grown ±vanillate for 24 h; right panel, cell length distribution at different timepoints during vanillate depletion. In the boxplots, boxes enclose the 25th and 75th percentile with the black line representing the mean, whiskers indicate the 10th and 90th percentile, and "+" indicates outliers. Numbers above each box indicate number of cells used for quantification (n) and mean ± standard deviation (SD). Cells were treated as in **b**. **\*\*p** < 0.001 in two-sided Student's t-test, in comparison to WT. Exact p-values and source data are provided in the Source data file. **d** Immunoblot analysis of CdbA-mCh accumulation. CdbA-mCh expression was induced with the indicated concentrations of vanillate for 24 h before cells were harvested. The WT/ Pnat_cdbA-mCh strain was used as a control for the native CdbA level. PilC was used as a loading control. Protein from the same number of cells was loaded per lane. Molecular mass marker is indicated on the left. Similar results were obtained in two independent experiments. Source data are provided as a Source Data file. **e** Immunoblot analysis of CdbA-mCh accumulation in the ΔcdbA/Pvan_cbdA-mCh strain during time course of vanillate depletion. PilC was used as loading control. Protein from the same number of cells was loaded per lane. Molecular mass marker is indicated on the left. Similar results were obtained in two independent experiments. Source data are provided as a Source Data file.

**CdbA is essential for viability while CdbB is dispensable**. We attempted to generate in-frame deletions in *cdbA* and *cdbB*. We readily obtained the in-frame deletion of *cdbB* and the mutant was indistinguishable from WT with respect to growth, cell morphology, motility and development (Fig. 4a–c, Supplementary Fig. 3a). By contrast, we were unable to generate an in-frame deletion of *cdbA* (Δ*cdbA*). We, therefore, constructed two merodiploid derivatives of WT in which mCherry (mCh)-tagged

CdbA was expressed ectopically. In one strain, *cdbA-mCh* was expressed from the native *cdbA* promoter in a single copy from the Mx8 *attB* site (WT/Pnat_*cdbA-mCh*) and in the second, *cdbA-mCh* was expressed from the vanillate-inducible promoter in a single copy from the MXAN_18/19 locus (WT/Pvan_*cdbA-mCh*). In immunoblots, CdbA-mCh accumulated in both strains; however, even at the highest concentration of vanillate (500 μM), the level of CdbA-mCh was ~9-fold lower when expressed from the vanillate-inducible promoter than when expressed from the native promoter (Fig. 4d). In the absence of vanillate, CdbA-mCh was undetectable by immunoblotting in the WT/Pvan_*cdbA-mCh* strain verifying that Pvan tightly regulates *cdbA-mCh* expression (Fig. 4d). Using the WT/Pvan_*cdbA-mCh* strain, we successfully generated an in-frame deletion of *cdbA* at the native locus in the presence of 500 μM vanillate (Δ*cdbA*/Pvan_*cdbA-mCh*). This strain grew in the presence of 500 μM vanillate but at a reduced rate compared to WT (Fig. 4a) while a derivative of this strain in which the Pnat_*cdbA-mCh* construct was also integrated at the Mx8 *attB* site grew like WT in the absence of vanillate (Supplementary Fig. 3b, c). We conclude that the CdbA-mCh protein is fully active and that the reduced growth rate of the Δ*cdbA*/Pvan_*cdbA-mCh* strain is likely caused by the lower accumulation of CdbA-mCh.

Removal of vanillate caused growth arrest of Δ*cdbA*/Pvan_*cdbA-mCh* cells after 24 h correlating with the earliest time-point at which CdbA-mCh was no longer detectable by immunoblotting and also caused a four-log defect in plating efficiency (Fig. 4a, e). Cells of the CdbA-mCh depletion strain were slightly longer than WT cells in the presence of vanillate, while cells grown under CdbA-mCh depleting conditions for 24 h were highly filamentous (Fig. 4b, c). These cells still formed constrictions at midcell but failed to complete division and eventually lysed (Fig. 4b). We conclude that CdbA is essential for viability and that lack of CdbA causes a cell division defect.

To investigate the relationship between CdbA and CdbB, we generated a Δ*cdbAB* double deletion in the strain containing the Pvan_*cdbA-mCh* construct (Δ*cdbAB*/Pvan_*cdbA-mCh*). In the presence as well as in the absence of vanillate, cells of this strain had an even more significant growth and cell division defect than the Δ*cdbA*/Pvan_*cdbA-mCh* cells (Fig. 4a–c) suggesting that CdbB can partially substitute CdbA. Interestingly, CdbA and CdbB interact in a bacterial two-hybrid assay suggesting that they can form heterooligomers (Supplementary Fig. 3d). Altogether, these observations indicate that CdbA and CdbB have similar functions. However, CdbA alone is required and sufficient for viability while CdbB is neither required nor sufficient for viability.

**CdbA binds the nucleoid globally and sequence-specifically.** Because CdbA is a member of the RHH family of DNA binding proteins, we asked whether CdbA binds DNA. In fluorescence microscopy analyses of WT/Pnat_*cdbA-mCh* cells, CdbA-mCh colocalized with the DAPI-stained nucleoid in *M. xanthus* (Fig. 5a). Similarly, CdbA-mCh colocalized with the nucleoid in *E. coli* (Fig. 5b) supporting that CdbA binds DNA in vivo.

We used ChIP-seq to identify CdbA binding sites at a genome-wide scale in *M. xanthus* using a strain in which *cdbA* at the native locus was replaced with an allele encoding a C-terminally 3×FLAG-tagged CdbA variant. This strain grew like WT, indicating that the protein was fully functional (Supplementary Fig. 4a). As a positive control, we used a strain expressing 3×FLAG-tagged ParB from the native locus and as a negative control, WT without a FLAG-tagged protein. In the two replicates, the WT control only showed a few peaks enriched ≥4-fold over input (Fig. 5c), while the ParB-3×FLAG control showed a broad peak in the region containing the *parS* sites

(Fig. 5c). The ChIP-seq peaks for CdbA-3×FLAG revealed broad occupancy over the *M. xanthus* chromosome (Fig. 5c). For both 3×FLAG-tagged proteins, the ChIP-seq peaks were strongly correlated between the two replicates while no correlation was observed for the WT control (Fig. 5c).

We detected 569 peaks with ≥4-fold enrichment over input for CdbA in intergenic regions and within structural genes (Fig. 5c, Supplementary Data 2). To experimentally verify the ChIP-seq results, we performed electrophoretic mobility shift assays (EMSAs) with purified CdbA-His$_6$ and 250 bp fragments corresponding to peak rank #1, #8, #161, and a fragment with no peak as a control. CdbA bound to the peak-fragments with an affinity that decreased with decreasing peak rank ($K_d$ from ~0.4 to ~0.9 μM) and the fragment with no peak had significantly lower affinity ($K_d$ of ~4 μM) (Fig. 6a) suggesting that CdbA binds with some sequence specificity and non-specifically at high concentration. Consistently, at the highest concentration of CdbA-His$_6$ (4 μM) the shifted complexes tended to run higher in the gels than at lower concentrations supporting that more than one CdbA molecule would bind to a DNA fragment (Fig. 6a).

To identify the CdbA consensus DNA binding sequence, we searched the 100-bp sequences centered around each peak summit using the MEME-ChIP web tool. Using the top 100 peak sequences, we identified a direct repeat motif separated by four bp, close to the peaks summits (Fig. 6b), while only half of this motif was detected when we used the top 500 peak sequences (Fig. 6b).

To analyze the importance of the direct repeat, we performed EMSAs with a DNA fragment from peak rank #1 containing the WT sequence or fragments in which the two most conserved bp had been mutated in one or both repeats (Fig. 6c). While the WT sequence had a $K_d$ of ~0.4 μM, the fragment with one repeat mutated had a $K_d$ of ~1.1 μM and the fragment with two repeats mutated had a $K_d$ of ~2.5 μM, close to that for unspecific binding by CdbA ($K_d$ ~ 4 μM; Fig. 6a, c). These observations support that the identified motif is important for sequence-specific DNA binding by CdbA and that its conservation dictates CdbA affinity for DNA. The repeat fits well to the model of CdbA DNA binding in which the two DNA binding regions are separated by one helical turn. Using a hidden Markov model, we searched the *M. xanthus* genome for the presence of the direct repeat motif and detected 8917 sequences with a *p*-value < 0.00077. Out of the 569 ChIP-seq peaks, 296 (52%) have at least one identifiable direct repeat motif present in a distance ±50bp from the peak summit (Supplementary Fig. 4c, Supplementary Data 2).

The model of DNA bound tetrameric CdbA, predicts that CdbA bends DNA when bound to the direct repeat (Fig. 2d). To test this prediction, we performed a circular permutation analysis using 250 bp fragments from peak rank #1 in which the direct repeat was systematically placed at different locations along the 250 bp. All DNA fragments showed the same migration with no CdbA bound (Supplementary Fig. 4d) while CdbA binding induced a more pronounced shift in the DNA fragment containing the motif towards the center of the fragment in comparison to fragments in which the binding site was positioned towards the end, demonstrating that CdbA bends DNA upon binding (Supplementary Fig. 4d).

Using quantitative immunoblot analysis, we determined the number of CdbA monomers per cell to ~7000 corresponding to a concentration of 2.2 μM for the tetramer supporting that CdbA is able to broadly occupy the *M. xanthus* genome (Supplementary Fig. 4e). We conclude that CdbA binds to >500 sites along the *M. xanthus* chromosome, that CdbA likely binds DNA as a tetramer with a consensus sequence consisting of a direct repeat and causes DNA bending upon binding.

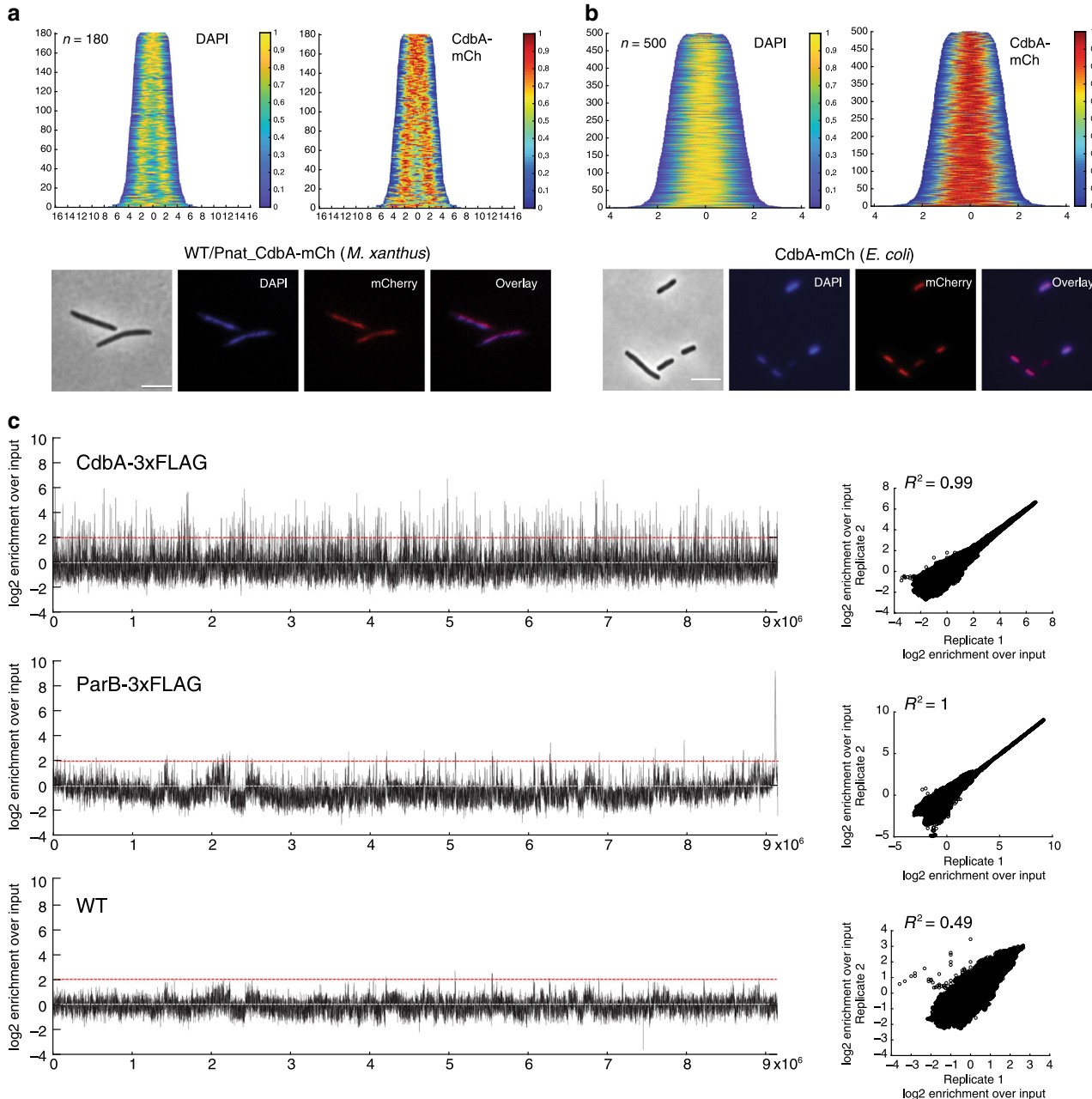

**Fig. 5 CdbA binds multiple sites across the M. xanthus genome. a, b** CdbA-mCh localization and DAPI staining of the nucleoid in WT *M. xanthus and E. coli*. In the demographs, cells were sorted according to cell length and fluorescence profiles of individual cells stacked with the shortest cell at the top and the longest at the bottom. *n* is indicated for each strain. Below, images of representative cells. Scale bars, 5 μm. **c** Genome-wide ChIP-seq profiles of CdbA-3×FLAG-bound and ParB-3×FLAG-bound regions on the *M. xanthus* chromosome. WT was used as a negative control. The log2 enrichment ratio was calculated from IP DNA and input DNA and plotted against location on the 9.28 Mb *M. xanthus* chromosome for one replicate. Grey line indicates no difference between sample and input. Red line indicates 4-fold difference (log2 = 2) set as a significance threshold for peaks. Right panels, scatter plots show correlation between two replicates for each strain. The log2 enrichment over input for replicate 1 is plotted against log2 enrichment over input for replicate 2 for each genomic position. The correlation coefficient, $R^2$, is indicated for each strain.

**CdbA depletion causes only minor changes in transcription.** 198 of the CdbA ChIP-seq peaks (~35%) mapped to intergenic regions, and among the top 100 peaks, 47% were intergenic (Supplementary Data 2) while less than 10% of the *M. xanthus* genome is intergenic[42]. Genes downstream from the 198 intergenic peaks encode proteins annotated as hypothetical or involved in various cellular functions and no specific functional category was enriched (Supplementary Data 2). Cell division defects resulting in cell filamentation can be caused by defects in divisome assembly[29,30,43,44], peptidoglycan synthesis[29], DNA

replication as well as DNA damage[45–49], chromosome segregation or chromosome organization[50,51]. Because CdbA-depletion caused cell filamentation with incomplete cell division, we searched all 569 peaks for genes known to be involved in such functions and with a peak in the predicted promoter region but did not identify any (Supplementary Data 2).

To determine whether CdbA functions as a transcriptional regulator, we performed RT-qPCR analysis on total RNA isolated from WT and CdbA-mCh-depleted cells. Among the nine genes tested, which all had a ChIP-seq peak in the predicted promoter

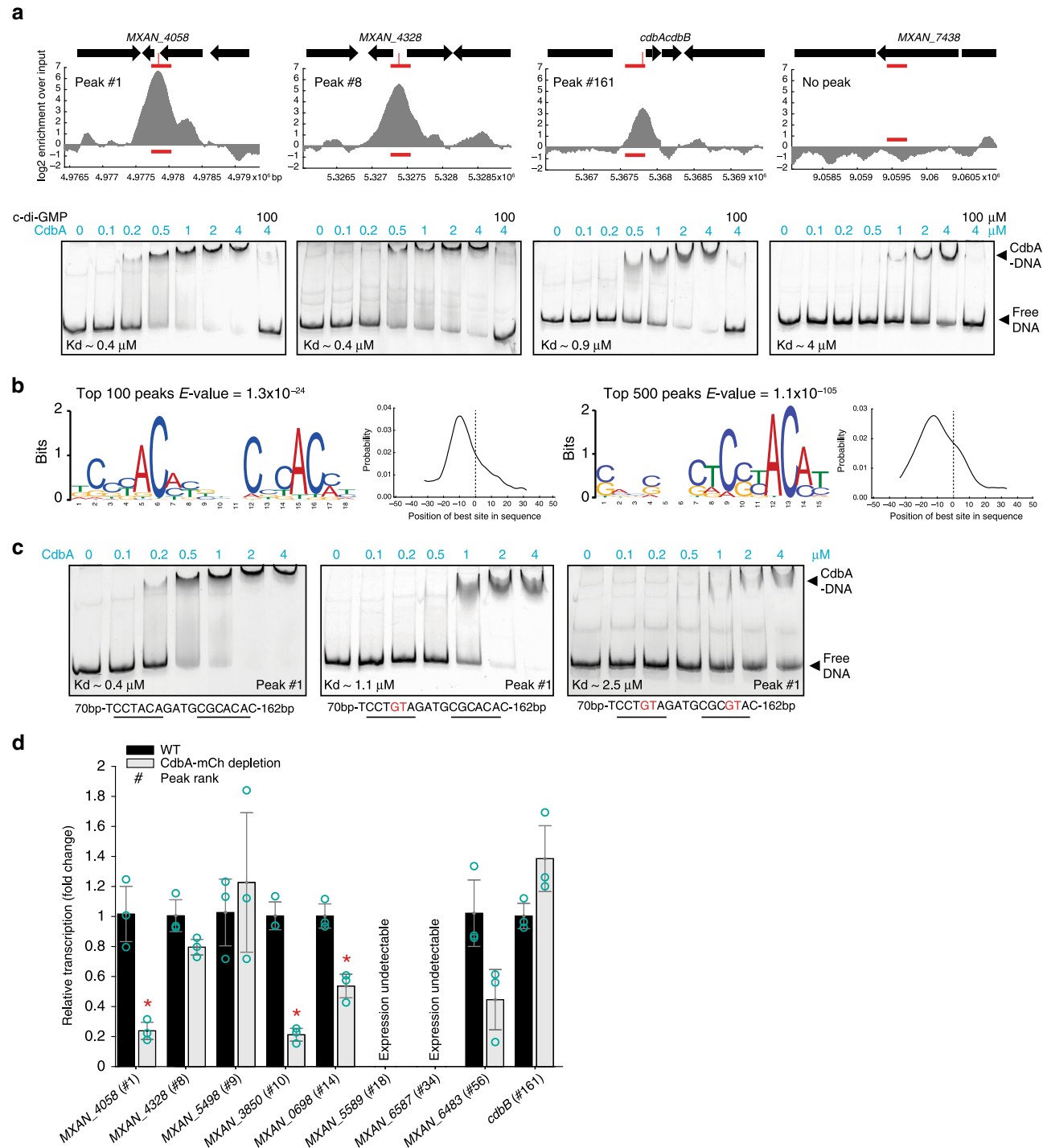

region, three showed minor but significant decreases in transcription upon CdbA-mCh depletion, but this effect did not correlate with peak rank; four were unaffected by CdbA depletion and two were not detectably expressed under the conditions tested (which were identical to those under which the ChIP-seq experiment was performed) (Fig. 6d). Thus, even though CdbA binds in the promoter regions of genes, it does not function as a classical transcriptional regulator consistent with the observation that most ChIP-seq peaks are within structural genes.

**c-di-GMP inhibits DNA binding by CdbA**. As noted, the regions in CdbA involved in c-di-GMP binding overlap with the regions predicted to be important for DNA binding (Figs. 2d and

3e). We analyzed whether c-di-GMP affects DNA binding by CdbA using a DNA fragment that covers 250 bp upstream of the *cdbA* translational start codon. CdbA-His$_6$ binds this fragment with a $K_d$ of 0.9 μM (Fig. 6a). Increasing c-di-GMP concentrations reduced CdbA DNA binding and at 16 μM, DNA binding was completely abolished (Fig. 7a). Because CdbA binds c-di-GMP with a $K_d$ of 83 nM while c-di-GMP only reduced CdbA DNA binding in the μM range, we performed an order of addition experiments in which we first added c-di-GMP to CdbA followed by DNA or vice versa. Independently of the order of addition, c-di-GMP had the same inhibitory effect on CdbA DNA binding (Supplementary Fig. 5). This effect was specific to c-di-GMP as none of five other tested nucleotides had an effect on

**Fig. 6 CdbA binds DNA in a moderately sequence-specific manner. a** CdbA binds DNA with sequence specificity. Upper panels, zoom in on ChIP-seq results for selected DNA fragments from different peaks sorted by peak rank and shown as log2 enrichment over input for one replicate. ORFs are shown as black arrows. Horizontal red line indicates 250 bp fragments used for EMSA analysis with position of peak summit indicated with a vertical red line. Lower panels, EMSA analysis of CdbA binding to DNA fragments shown in upper panels. Concentration of CdbA added is indicated. c-di-GMP was added to a final concentration of 100 µM in the last lane, 10 min prior addition of DNA probe. $K_d$ was calculated based on two replicates. Source data are provided as a Source Data file. **b** MEME-ChIP results for CdbA binding motif search. Left panels, motif identified based on top 100 peaks and the probability distribution of the identified motif occurrence across the input sequences in which position 0 corresponds to peak summits; right panels, as in the left panels except that the top 500 peaks were analyzed. For both analysis 100 bp (peak summit ±50 bp) were used as input sequences for the MEME-ChIP analysis. **c** EMSA analysis of CdbA binding to WT and mutant DNA fragments of *MXAN_4058* promoter. Below each image, the sequence of the WT binding motif and the two mutant variants are indicated with substituted bp indicated in red. Experiments were done as in (a) and $K_d$ calculated based on two replicates. Source data are provided as a Source Data file. **d** RT-qPCR analysis of gene expression in WT and a strain depleted for CdbA-mCh for 24 h. Peak rank is indicated. Transcript levels are shown as mean ± SD from three biological replicates with each three technical replicates relatively to WT. *$p <$ 0.05 in two-sided Student's *t*-test, between WT and CdbA-mCh depletion. Individual data points are in green–gray. Exact *p*-values and source data are provided in the Source data file.

CdbA DNA binding (Fig. 7b). Similarly, c-di-GMP abolished binding to the other DNA fragments tested in EMSAs (Fig. 6a).

CdbA$^{K8A/S10A}$-His$_6$ and CdbA$^{R27A/R30A}$-His$_6$, which do not detectably bind c-di-GMP (Fig. 1c, d), also failed to bind the *cdbAB* promoter in EMSAs (Fig. 7c). These observations are in agreement with a model in which the same regions in CdbA are important for DNA binding and c-di-GMP binding and that binding of these two ligands is mutually exclusive.

To determine whether DNA binding and/or c-di-GMP binding by CdbA is important in vivo, we generated three strains in which (A) the native *cdbA* gene was deleted (Δ*cdbA*), (B) CdbA$^{WT}$-mCh was expressed from the vanillate-inducible promoter, and (C) CdbA$^{WT}$-mCh, CdbA$^{K8A/S10A}$-mCh or CdbA$^{R27A/R30A}$-mCh were expressed from the native promoter from the Mx8 *attB* site. Upon removal of vanillate, these three strains accumulate CdbA$^{WT}$-mCh, CdbA$^{K8A/S10A}$-mCh or CdbA$^{R27A/R30A}$-mCh, respectively (Fig. 7d). While the strain accumulating CdbA$^{WT}$-mCh grew as WT (Fig. 7e), the two strains accumulating CdbA$^{K8A/S10A}$-mCh or CdbA$^{R27A/R30A}$-mCh had a growth defect (Fig. 7e) and became filamentous (Fig. 7f) demonstrating that the variants are inactive in vivo and suggesting that DNA and/or c-di-GMP binding is essential for CdbA function in vivo. The two variants also failed to localize over the nucleoid in *M. xanthus* (Fig. 7f), confirming the in vitro result that they have a DNA binding defect (Fig. 7c).

**CdbA depletion affects chromosome organization and segregation.** Having ruled out that CdbA functions as a classical transcription factor and inspired by its abundance and global occupancy over the *M. xanthus* chromosome and essentiality, we hypothesized that CdbA could be a nucleoid-associated protein (NAP). We, therefore, analyzed nucleoid organization and segregation in CdbA-mCh-depleted cells using DAPI staining to assess nucleoid organization and a ParB-YFP fluorescent fusion as a marker for the origin and to assess chromosome organization and segregation[31].

In the presence of vanillate, WT and the CdbA-mCh depletion strain showed similar DAPI staining patterns with a single nucleoid in short cells and two fully replicated and segregated nucleoids in longer cells (Fig. 8a). As previously observed[31–33], most WT cells had two ParB-YFP signals at 25 and 75% of the cell length while slightly more cells of the CdbA-mCh depletion strain had more than two ParB signals (Fig. 8a, b). Also, the distance from the ParB-YFP clusters to the nearest cell pole was slightly but significantly longer than in WT (2.2 ± 0.8 vs 1.8 ± 0.6 µm) (Fig. 8c). As a control for the CdbA-mCh depletion strain in the absence of vanillate, WT cells were artificially elongated with 8 h of cephalexin treatment, which inhibits cell division without affecting chromosome replication, organization and segregation[29]

(Fig. 8a, b). As reported[29], cephalexin treated WT cells had a regular distribution of nucleoids and ParB-YFP signals along the cell length. By contrast, in the CdbA-mCh depleted cells, nucleoids appeared more condensed, less well separated and mostly localizing in the center of the long cells (Fig. 8a, b, Supplementary Fig. 6a). Also, ParB-YFP foci were not regularly distributed along the cell length but clustered in the center of cells paralleling the localization of the nucleoid (Fig. 8a, b, Supplementary Fig. 6a). Consistently, in these cells, the distance from the ParB-YFP clusters to the nearest cell pole was significantly larger than in cephalexin treated cells (5.3 ± 2.4 vs 3.7 ± 2.5 µm) (Fig. 8c, Supplementary Fig. 6b). This phenotype is different from Δ*bacNOP* and Δ*padC* strains where localization of ParB-YFP foci was also affected but no cellular filamentation was observed[33].

In time-lapse recordings of non-motile cells, we observed that segregation of ParB clusters was significantly slower and more erratic after CdbA-mCh depletion compared to that in WT[30–33] (Supplementary Fig. 6c). Despite the abnormal localization of ParB-YFP clusters and the abnormal morphology and localization of the nucleoid, the number of ParB-YFP clusters per cell length increased similarly for cephalexin treated WT and the CdbA-mCh depletion strain (Fig. 8a) suggesting that DNA replication is unaffected by depletion of CdbA-mCh. These observations demonstrate that CdbA is important for chromosome organization and segregation in agreement with the idea that CdbA is a NAP.

Because DNA and c-di-GMP binding by CdbA is mutually exclusive in vitro, we speculated that if the essential function of CdbA lies in its NAP activity, then changing the cellular concentration of c-di-GMP would affect CdbA DNA binding and, therefore, chromosome organization and segregation. To test this hypothesis, we analyzed cell length distribution and nucleoid morphology in a strain that accumulates the heterologous DGC DgcA of *Caulobacter crescentus*, and which accumulates c-di-GMP at a 7-fold higher level than WT and in a strain that accumulates the heterologous PDE PA5295 of *Pseudomonas aeruginosa* and which accumulates c-di-GMP at a 2-fold lower level than WT[26]. These strains grow as WT and we neither observed an effect on cell length nor on nucleoid morphology (Supplementary Fig. 7a, b).

**Discussion**

Here, we report the identification of the DNA binding RHH protein CdbA in *M. xanthus*, a previously undescribed type of c-di-GMP receptor, that globally binds the *M. xanthus* chromosome and is essential for viability. Cells depleted for CdbA have severe defects in chromosome organization and segregation and are impaired in cell division and highly filamentous. Taken together our data support a model whereby CdbA is an essential

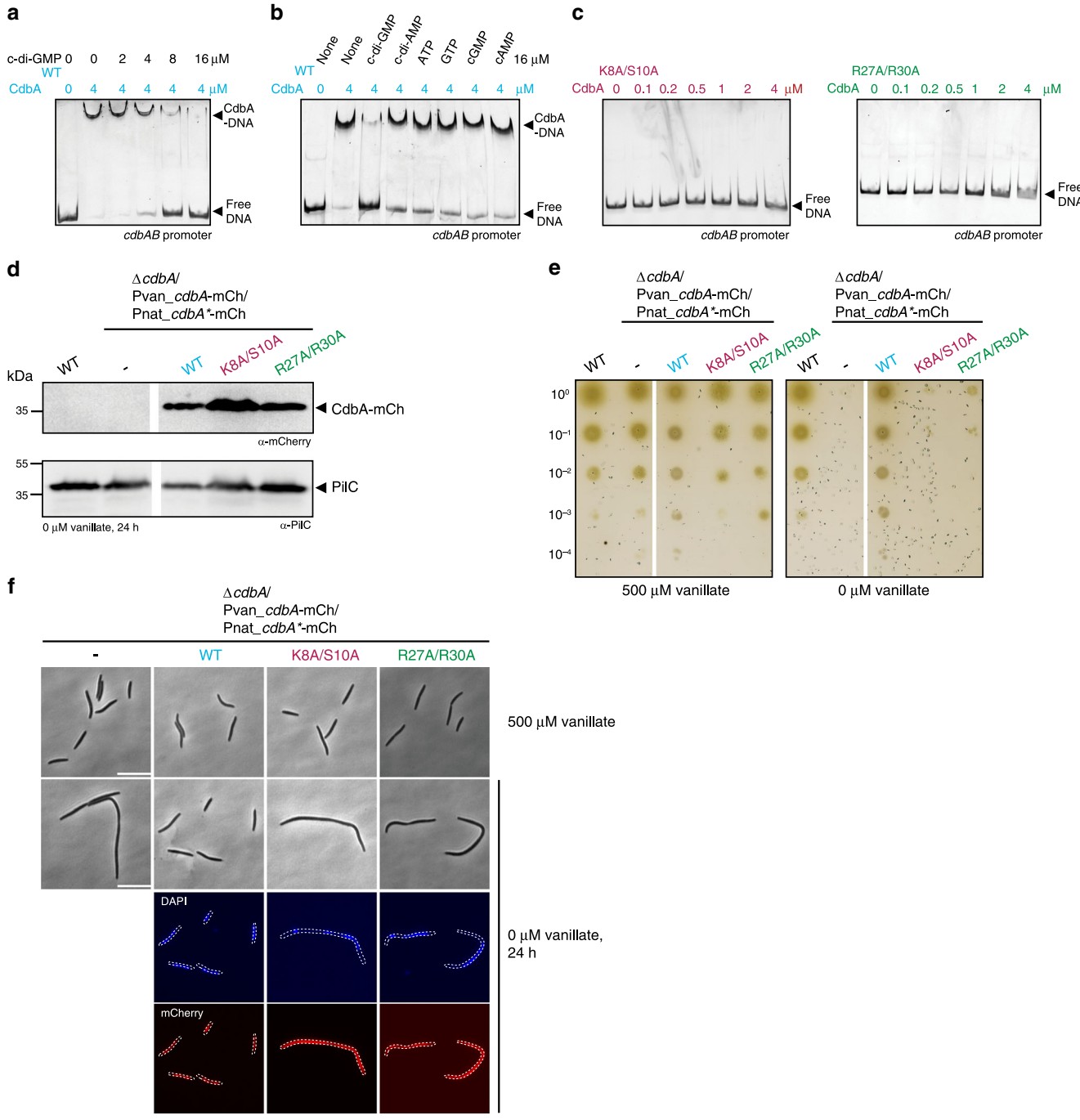

**Fig. 7 C-di-GMP and DNA binding is mutually exclusive and essential for CdbA function in vivo. a** EMSA experiment with CdbA in the presence of c-di-GMP. The fragment used covers the *cdbAB* promoter (see also Fig. 5a). 4 µM CdbA was incubated with indicated concentrations of c-di-GMP for 10 min. prior to addition of DNA fragment. Similar results were obtained in two independent experiments. Source data are provided as a Source Data file. **b** EMSA experiment with CdbA in the presence of different nucleotides. Experiment was done as in **a** and a concentration of 16 µM for all nucleotides was used. The same results were obtained in two independent experiments. Source data are provided as a Source Data file. **c** EMSA experiment with CdbA^K8A/S10A and CdbA^R27A/R30A. Experiments were done as in **a**; CdbA concentrations are indicated. The same results were obtained in two independent experiments. Source data are provided as a Source Data file. **d** Immunoblot analysis of accumulation of CdbA-mCh variants in indicated strains after 24 h of vanillate depletion. PilC was used as a loading control. Protein from the same number of cells was loaded per lane. For both proteins, samples are from the same blot and lanes were removed for presentation purposes. Molecular mass marker is indicated on the left. Similar results were obtained in two independent experiments. Source data are provided as a Source Data file. **e** Growth of indicated strains on solid surface in the presence and absence of vanillate driving synthesis of CdbA^WT-mCh. Plates were incubated for 4 days. All the strains were spotted on the same plates. Similar results were obtained in two independent experiments. Source data are provided as a Source Data file. **f** CdbA-mCh localization and DAPI staining of the nucleoid in representative cells of indicated genotypes grown in the presence and absence of vanillate. In the absence of vanillate, cells were imaged 24 h after vanillate removal. Scale bars, 10 µm. Similar results were obtained in two independent experiments.

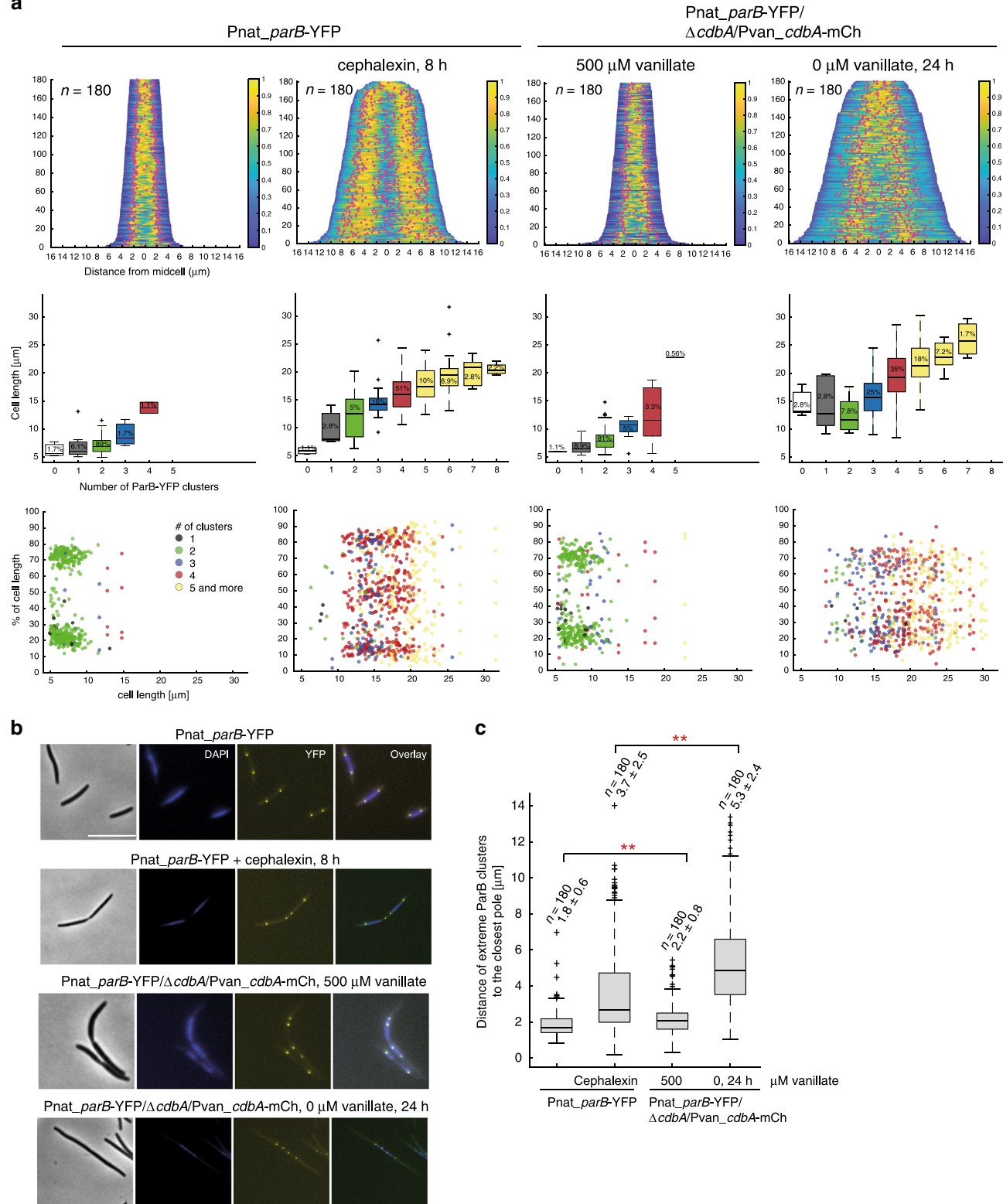

and ligand-regulated NAP, the activity of which is modulated by c-di-GMP binding.

Generally, NAPs are small, abundant and dissimilar proteins that bind DNA with moderate sequence specificity causing bending, wrapping or bridging of DNA and, typically, only regulate transcription on a small scale[52,53]. The moderate effects on gene expression distinguishes NAPs from classical transcriptional

regulators whose target genes are normally more dramatically regulated. For example, Fis, a NAP of *Escherichia coli*, recognizes ~900 regions across the genome, bends DNA, and generally only causes minor changes in transcription[54]. While Fis is non-essential for viability, in its absence, cells are elongated, have chromosome segregation defects, and less condensed nucleoids compared to WT[55]. H-NS, a NAP and global repressor of

**Fig. 8 Lack of CdbA affects chromosome organization and segregation. a** CdbA is important for chromosome organization and segregation. First row, demographs with overlay of DAPI-stained nucleoids and ParB-YFP localization (red circles) in strains of indicated genotypes and in the presence and absence of vanillate and cephalexin as indicated. In the absence of vanillate, cells were imaged 24 h after vanillate removal; cephalexin was added for 8 h. Second row, box plot showing number of ParB-YFP foci per cell as a function of cell length. In the boxplots, boxes enclose the 25th and 75th percentile with the black line representing the mean, whiskers indicate the 10th and 90th percentile, and "+" indicates outliers. Numbers in bars indicate the percentage of analyzed cells having the indicated number of ParB-YFP foci. Third row, scatter plots of position of ParB-YFP clusters along the long axis of cell in different strains as a function of cell length. $n = 180$ for each strain and for each analysis. Source data are provided as a Source Data file. **b** Images of representative cells used for the analysis in **a**. Scale bar, 10 μm. **c** Box plot showing distance between ParB-YFP clusters and the closest pole as a function of cell length for cells with at least two ParB-YFP clusters. The same 180 cells as in **a** were analyzed and only the ParB-YFP clusters closest to a pole were used in the analysis. Box plot as in **a**. Numbers above each box indicate number of cells used for quantification (n) and mean ± SD. **$p < 0.001$ in two-sided Student's t-test. Exact p-values and source data are provided in the Source data file.

transcription in enterobacteria, shows sequence-specific DNA binding to ~500 AT-rich regions and affects chromosome organization and transcription by forming DNA loops[56]. GapR in *C. crescentus* is an essential NAP that recognizes AT-rich sequences, binds to ~600 regions across the genome preferentially associating with intergenic regions. GapR depletion causes formation of filamentous—and minicells but does not cause major changes in transcription[57,58]. By comparison CdbA binds DNA with some sequence specificity and a $K_d$ of 0.4–4.0 μM, which is significantly lower than for other RHH proteins that have $K_d$s in the range of 10-200 nM[59–61]. Forty-seven percent of the top 100 ChIP-seq peaks are in intergenic regions; however, RT-qPCR experiments comparing transcription in WT and CdbA-depleted cells either revealed modest effect, no effect or no expression of genes with a CdbA binding site in the promoter region suggesting that CdbA does not function as a classical transcriptional regulator. Also, the putative functions of proteins encoded by genes with a ChIP-seq peak in the promoter region did not show enrichment of functional categories as would be expected for a classical transcriptional regulator. Moreover, these genes did not include those encoding proteins involved in cell growth, cell division, chromosome organization and/or segregation that could explain the chromosome and cell division defects in cells depleted of CdbA. Additionally, CdbA binds the 9.28 Mb *M. xanthus* genome at >500 sites, bends DNA upon binding, is highly abundant with ~1700 tetramers per cell, and is essential. Altogether, these observations are consistent with the notion that CdbA is a NAP with an essential function in chromosome organization and segregation. According to this model, the primary function of CdbA is that of a NAP to assist in organizing the chromosome and support chromosome segregation while the effects on transcription and cell division are likely secondary to the defect in chromosome organization. In contrast to other NAPs, the CdbA consensus binding site is GC-rich. We speculate that this might be explained not only by the GC-rich nature of *M. xanthus* genome (69% GC content) but also by the mutually exclusive binding of DNA and c-di-GMP potentially allowing the guanine bases of c-di-GMP and GC-rich DNA fragment to compete for the same binding site on CdbA (see below). Other NAPs are typically regulated by abundance[52,53]; whether CdbA is also regulated by abundance is not known; however, our data support the notion that CdbA's DNA binding activity is regulated by c-di-GMP, i.e., CdbA is a ligand-regulated NAP.

In vitro c-di-GMP binding by the CdbA tetramer induces conformational changes. C-di-GMP and DNA binding are mutually exclusive and the two ligands appear to compete for the same binding interfaces on CdbA. In other RHH proteins the β-sheet provides specificity in DNA binding, while the loop region between α-helix1 and α-helix2 together with the N-terminus of α-helix2 make DNA backbone contacts and increases the affinity for DNA[37]. Precisely these two regions show a modified response upon addition of c-di-GMP in the HDX experiment. The 1:2 ratio

of c-di-GMP:CdbA, HDX data of binding interfaces, and our structure of a tetramer formed from two RHH dimers allow the postulation of two models for c-di-GMP binding. The first model would place one c-di-GMP molecule directly over each β-sheet in the tetramer; this singular dinucleotide would adopt either the extended conformation observed in EAL PDEs[62] or the clamp-like conformations adopted for example in the HD-GYP domain protein PmGH[63]. In this model, the alterations in HDX of the α-helix1-2 loop could be explained by a change in the relative orientation of the two CdbA dimers in the tetramer upon c-di-GMP binding. A second, intriguing possibility is that the unique dimer:dimer pocket formed at the center of the tetramer acts to bind the commonly encountered intercalated dimer form of c-di-GMP[64], and the two β-sheets flex inward to explain their implication in the HDX data. Flexation would also allow the helical turn (residues 29–34) in the center of the tetramer to recognize the c-di-GMP phosphate group in the same manner that RHH proteins use to bind the DNA backbone. The tetramer interface displays different hydrophobic packing between the two crystal forms we observed here, suggesting that it may have inherent plasticity that allows it to undergo this change. In total, the first model would rely on steric blockage to enable c-di-GMP to license CdbA release from DNA, whereas the second model could modulate DNA-binding via c-di-GMP alteration of the β-sheet:β-sheet distance. RHH domains are found as stand-alone RHH domains or together with other domains[37]. Ligand binding and modulation of DNA binding by other RHH proteins has only been demonstrated for RHH proteins with additional domains and the ligands bind to these extra domains[65,66]. Thus, CdbA is the first RHH protein that binds a ligand (other than DNA) via the RHH domain. In *P. aeruginosa* AmrZ, one of the best studied RHH proteins, is involved in regulating the c-di-GMP level[67]. However, we did not observe c-di-GMP binding to AmrZ (Supplementary Fig. 8) documenting that c-di-GMP binding is not a general feature of RHH proteins.

In contrast to CdbA, other c-di-GMP binding transcriptional regulators use distinct domain for c-di-GMP and DNA binding. C-di-GMP may stimulate DNA binding by stimulating oligomer formation as is the case for VpsT of *Vibrio cholerae*[18], BldD of *Streptomyces coelicolor*[19] and CuxR of *Sinorhizobium meliloti*[20]. Alternatively, upon c-di-GMP binding, FleQ of *Pseudomonas aeruginosa* undergoes a conformational change causing reduced DNA binding[21].

In the absence of DNA, CdbA binds c-di-GMP with a $K_d \sim$ 83 nM; and CdbA binds DNA fragments with $K_d \sim$ 0.4–4 μM (depending on the fragment used). Nevertheless, in vitro c-di-GMP only reduced CdbA DNA binding when added in the μM range indicating the complexity of this interplay; there is a possibility that DNA-binding changes CdbA structure/dynamics such that c-di-GMP binding is less favorable than when DNA is absent (i.e., this is not represented by a simple competition model, and can be enacted by an extensive interface for DNA binding

that is more than just the two β-sheets). The estimated c-di-GMP concentration in *M. xanthus* cells in rich medium is ~1.4 ± 0.5 μM[26], suggesting that c-di-GMP binding by CdbA should be relevant in vivo. However, strains in which the c-di-GMP level was artificially increased or decreased 7- or 2-fold, respectively, have no defects in growth, chromosome organization/segregation or cell division. Based on these observations and the observation that inactivation of CdbA has a dramatic effect on viability, we speculate that alteration in c-di-GMP may not elicit an all-or-none response with respect to CdbA DNA binding but rather a graded response in which c-di-GMP binding to CdbA modulates or fine-tunes CdbA DNA binding in vivo. Alternatively, exposure of CdbA to c-di-GMP is controlled by complex formation with specific DGCs/PDEs creating a local c-di-GMP pool. In this scenario, manipulating the global c-di-GMP level would not affect CdbA-DNA binding. Of note, because the ~10-fold increase in c-di-GMP during development is essential for fruiting body formation and sporulation[27] and spores are diploid[68] it is interesting to speculate that CdbA could participate in development specific chromosome reorganization and cell division inhibition. In the future, it will be of interest to identify other conditions that cause significant changes in the level of c-di-GMP.

In conclusion, we have identified the NAP CdbA in *M. xanthus* and our data support that its activity is modulated by the second messenger c-di-GMP, thus, revealing a link between c-di-GMP signaling and chromosome biology.

## Methods

**M. xanthus and E.coli strains and growth**. *M. xanthus* strains used in this study are all derivatives of the WT DK1622[69]. *M. xanthus* strains, plasmids and oligonucleotides used are listed in Supplementary Table 2, 3 and 4. *M. xanthus* cells were grown in liquid 1% CTT medium or on 1% CTT/1.5% agar plates at 32 °C[70]. Kanamycin and oxytetracycline were added to *M. xanthus* cells at concentrations of 40 μg ml⁻¹ or 10 μg ml⁻¹, respectively. Vanillate was added where indicated at the indicated concentrations. Growth was measured as an increase in OD at 550 nm. *E. coli* strains were grown in LB broth in the presence of relevant antibiotics[71]. All plasmids were propagated in *E. coli* Mach1 (ΔrecA1398 endA1 tonA Φ80ΔlacM15 ΔlacX74 hsdR($r_K^-$ $m_K^+$)).

**Motility and development assays**. For motility assays, cells were grown in CTT medium, harvested, and resuspended in 1% CTT to a calculated density of $7 \times 10^9$ cells ml⁻¹. Five-microliter aliquots of cell suspensions were placed on 0.5% and 1.5% agar supplemented with 0.5% CTT and incubated at 32 °C. After 24 h, colony edges were observed using a Leica MZ8 stereomicroscope or a Leica IMB/E inverted microscope and visualized using Leica DFC280 and DFC350FX charge-coupled-device cameras, respectively. For development assay, cells were harvested and resuspended in MC7 buffer (10 mM MOPS pH 7.0, 1 mM CaCl₂) to a calculated density of $7 \times 10^9$ cells ml⁻¹. Twenty microliter aliquots of cells were placed on TPM agar (10 mM Tris-HCl pH 7.6, 1 mM K₂HPO₄/KH₂PO₄ pH 7.6, 8 mM MgSO₄); for development in submerged culture, 50 μl of the cell suspension were mixed with 350 μl MC7 buffer and placed in an 18 mm diameter microtiter dish. Cells were visualized at the indicated time points using a Leica MZ8 stereomicroscope or a Leica IMB/E inverted microscope and imaged using Leica DFC280 and DFC350FX CCD cameras, respectively. Sporulation levels were determined after development for 120 h in submerged culture as the number of sonication- and heat-resistant spores relative to WT.

**Fluorescence microscopy and live cell imaging**. Exponentially growing cells were transferred to slides containing a thin pad of 1.5% agarose (Cambrex) with TPM buffer (10 mM Tris-HCl pH 7.6, 1 mM KH₂PO₄ pH 7.6, 8 mM MgSO₄, 0.2% CTT, covered with a coverslip and imaged with a temperature-controlled Leica DMi8 inverted microscope. Phase contrast and fluorescence images were acquired using a Hamamatsu ORCA-flash V2 Digital CMOS camera. Cells in phase contrast images were automatically detected using Oufti software[72]. Fluorescence signals were identified and analyzed using a custom-made Matlab v2016b (MathWorks) script. *E. coli* cells were induced with 0.05 mM isopropyl-β-D-thiogalactopyranosid (IPTG) for 2 h and treated with 30 μg ml⁻¹ chloramphenicol for 30 min before DAPI staining. For DAPI staining, cells were incubated with 1 mg ml⁻¹ DAPI for 10 min at 32 °C prior to start of microscopy. Image processing was performed using Metamorph_ v 7.5 (Molecular Devices).

**Bacterial two-hybrid assay (BACTH)**. BACTH experiments were performed as described[73]. Briefly, full-length *cdbA* and *cdbB* were cloned into the appropriate

vectors to construct N-terminal and C-terminal fusions with the 25-kDa N-terminal or the 18-kDa C-terminal adenylate cyclase fragments. cAMP production was observed by the formation of blue color on LB agar supplemented with 80 μg ml⁻¹ 5-bromo-4-chloro-3-indolyl-β-d-galactopyranoside (X-Gal) and 0.25 mM IPTG.

**Operon mapping**. Total RNA was isolated using a Trizol (Sigma) extraction method. RNA was treated with DNase I (Invitrogene) and purified with the RNeasy kit (Qiagen). PCR analysis was used to confirm that the RNA was DNA free. One microgram of RNA was used to synthesize cDNA with the High capacity cDNA Archive kit (Applied Biosystems) using random hexamer primers. For the operon mapping, following primer pairs were used: "4361 qPCR forw"/"4361 qPCR rev" (fragment 1), "4362 qPCR forw"/"4362 qPCR rev" (fragment 2) and "4361 qPCR forw"/"4362 qPCR rev" (fragment 3). Genomic DNA and RNA were used as positive and negative controls, respectively.

**Real-time PCR**. Total RNA was isolated and cDNA was synthetized as described for operon mapping. qRT-PCR was performed in 25 μl reaction volume using SYBR green PCR master mix (Applied Biosystems) and 0.1 μM primers specific to the target gene in a 7500 Real Time PCR System (Applied Biosystems). Relative gene expression levels were calculated using the comparative Ct method. All experiments were done with three biological replicates each with three technical replicates.

**Immunoblot analysis**. Immunoblots were carried out as described[71]. Briefly, rabbit polyclonal α-FLAG (Rockland, dilution 1:2000), α-mCherry (Biovision, dilution 1:10000), α-PilC[74] (dilution 1:5000) antibodies were used together with horseradish-conjugated goat anti-rabbit immunoglobulin G (Sigma-Aldrich, dilution 1:10000) as secondary antibody. Blots were developed using Luminata Crescendo Western HRP Substrate (Millipore) and visualized using a LAS-4000 luminescent image analyzer (Fujifilm).

**Determination of protein copy number per cell**. Number of CdbA molecules per cell was determined using a quantitative immunoblot analysis. Exponentially growing WT culture was used to prepare cell lysates for immunoblot analysis. Different amounts of cell lysates and purified protein were separated on a 15% SDS-gel, transferred to a nitrocellulose membrane and probed with α-FLAG antibodies. Signal intensities of the lysate bands were quantified using Fiji[75] and compared against a standard curve generated from known amounts of His₆_CdbA_3×FLAG from the same Immunoblot. Experiment was performed in two independent biological replicates.

**Protein purification**. For DRaCALA, EMSA and determination of copy number proteins were expressed in *E. coli* Rosetta 2(DE3) growing in 2×TY medium at 37 °C using 0.5 mM IPTG induction for 3 h. His₆-tagged proteins were purified using Ni-NTA affinity purification. Briefly, cells were harvested by centrifugation (3500 × g, 20 min, 4 °C), resuspended in buffer A (50 mM Tris-HCl, 150 mM NaCl, 10 mM imidazole, 1 mM DTT, 10% glycerol, pH 8.0) and lysed using a French pressure cell. After centrifugation (1 h, 48,000 × g, 4 °C) lysates were loaded on a Ni-NTA agarose column (Qiagen) and washed with 20× column volume using buffer B (50 mM Tris-HCl, 150 mM NaCl, 20 mM imidazole, 1 mM DTT, 10% glycerol, pH 8.0). Proteins were eluted with buffer C with imidazole gradient (50 mM Tris-HCl, 150 mM NaCl, 50–500 mM imidazole, 1 mM DTT, 10% glycerol, pH 8.0). For ITC, analytical SEC and HDX proteins were essentially purified as described previously[76]. Briefly, *Escherichia coli* BL21 (DE3) cells carrying the expression plasmids were grown in lysogeny-broth medium supplemented with 50 μg ml⁻¹ kanamycin and 12.5 g l⁻¹ D(+)-lactose-monohydrate for 20 h at 30 °C under rigorous shaking. Cells were harvested by centrifugation (3500 × g, 20 min, 4 °C), resuspended in lysis buffer (20 mM of HEPES-Na pH 8.0, 250 mM NaCl, 20 mM MgCl₂, 20 mM KCl, 40 mM imidazole) and broken by one passage through the M-110L Microfluidizer (Microfluidics). After centrifugation (47,850 × g, 20 min, 4 °C), the supernatant was loaded on a 5 ml HisTrap column (GE Healthcare) equilibrated with 5-column volumes (CV) lysis buffer. After washing with 5 CV of lysis buffer, the proteins were eluted with 5 CV elution buffer (lysis buffer containing 500 mM imidazole). Proteins were concentrated (Amicon Ultracel-3K (Millipore)) and applied to size-exclusion chromatography (SEC) on a HiLoad 26/600 Superdex 200 pg column (GE Healthcare) equilibrated in SEC buffer (20 mM of HEPES-Na, pH 7.5, 200 mM NaCl, 20 mM MgCl₂, 20 mM KCl). Protein-containing fractions were pooled, again concentrated (Amicon Ultracel-3K (Millipore)), deep-frozen in liquid nitrogen and stored at −80 °C. For structure determination a single transformant was grown overnight in LB at 37 °C degrees and used to inoculate 1 liter of auto-induction medium[77] which was then incubated at 37 °C for three hours before the temperature was decreased to 18 °C for 18 h before cells were harvested by centrifugation. The resulting cell pellets were then homogenized in Buffer A (20 mM imidazole-HCl, pH 7.5, 400 mM NaCl) supplemented with 0.05% Tween20 and 100 μg ml⁻¹ lysozyme. Resuspended cells were lysed by sonication and the lysate was clarified by centrifugation at 48,000 × g for 60 min. Soluble proteins were loaded onto a 5 ml HisTrap FF column (GE Healthcare) pre-equilibrated in Buffer A, washed extensively with buffer A, and

then eluted in buffer B (400 mM imidazole-HCl, pH 7.5, 400 mM NaCl). Fractions containing pure recombinant CdbA were pooled and dialysed overnight in dialysis buffer (20 mM HEPES-NaOH, pH 7.0, 300 mM NaCl) at 4 °C.

**Analytical size-exclusion chromatography (SEC)**. Prior analytical SEC, 200 μM CdbA wild-type or its variants were incubated with or without 200 μM c-di-GMP for 1 minute at 25 °C. Analytical SEC was carried out using a Superdex S200 Increase 10/300 GL column (GE Healthcare) at 0.5 ml/min flow rate with SEC buffer at 4 °C. A standard curve for molecular mass determination was obtained using a mixture of thyroglobulin (669 kDa), ferritin (440 kDa), carboanhydrase (158 kDa), aldolase (75 kDa), ovalbumin (44 kDa), conalbumin (29 kDa), and RNase A (13.7 kDa).

**Electrophoretic mobility shift assay (EMSA)**. The HEX-labelled DNA fragments were generated by PCR. Reactions were carried out in 10× EMSA buffer (380 mM HEPES, 380 mM NaCl, 50 mM MgCl₂, 50% glycerol, 10 mM DTT, 15 ng μl⁻¹ poly (dI-dC) (Sigma-Aldrich) in the presence of indicated concentrations of CdbA_His₆ and 50 ng DNA probe. C-di-GMP was added at indicated concentrations and order of addition of c-di-GMP and DNA is indicated in figure legends. Reactions were incubated at 30 °C for 30 min. Samples were loaded on a 10% polyacrylamide gel, and electrophoresed in 1× TBE at 4 °C. The gel was pre-run in 1× TBE for 0.5 h. Gels were scanned for HEX signals on a Typhoon Scanner (GE Healthcare).

**Circular permutation analysis**. Circular permutated DNA fragments were generated with PCR. Gel shift was performed as for EMSA assay but instead of using fluorescence, signal was detected using GelRed dye.

**Structure determination**. Concentrated CdbA (28 mg ml⁻¹) was used in sitting drop crystallization screening experiments. Protein (1.8 μl) was mixed with an equal volume of crystallization condition. Initial protein crystal needles, unsuitable for X-ray diffraction experiments, were obtained in 0.8 M Na Formate, 25% w/v PEG 2000 MME, 0.1 M tri-sodium citrate, pH 6.5. This condition was selected to seed for novel crystallization conditions via a micromatrix seeding protocol[78]. Briefly, the crystals from this sitting drop were harvested and vortexed in 500 μl of the condition mother liquor to generate a crystal seed stock. New sitting drop screening was performed with drops comprised of 1.8 μl protein and 1.2 μl crystallization condition which were then supplemented with 0.6 μl of seed stock. This screening approach yielded a dramatic improvement on the initial screening with a large number of three dimensional protein crystals. Crystals of the dimeric form of CdbA were grown in 0.1 M MgCl₂, 0.1 M HEPES-NaOH, pH 7.0, 15% w/v PEG 4000 supplemented with seed stock. Crystals grown in this condition were cryoprotected by sequential addition of mother liquor supplemented with 25% (v/v) glycerol prior to flash freezing in liquid nitrogen. A heavy atom derivative of this protein crystal was prepared by sequential addition of mother liquor supplemented with 5 mM K₂PtCl₄. After 70 min, the derivatized crystals were cryoprotected and harvested as described for the native form. The tetrameric form of CdbA crystallized in 0.1 M sodium malonate dibasic monohydrate, 0.1 M HEPES-NaOH, pH 7.0, 30% w/v polyacrylic acid sodium salt 2100 and were cryoprotected in mother liquor supplemented with 20% w/v ethylene glycol. Diffraction data were collected at the Diamond Light Source in Oxford, UK. Data reduction and processing was achieved using Xia2[79]. The SAD and molecular replacement phases were calculated using scripts in the PHENIX suite of programs[80]. Protein structures were built/modified using COOT[81], with cycles of refinement in both PHENIX and PDB-REDO[82]. The low sequence identity (15–20%) with known RHH structures enforced us to use of de novo phasing, which was provided by SAD data collected at the LIII edge of a Platinum derivative of form one. The derivative gave a clear result (Pt modification of residue M17, FOM 0.36) and allowed tracing of residues 5–52.

**c-di-GMP capture compound experiments**. Experiments were performed as described[19,35]. Briefly, *M. xanthus* cultures were grown in liquid CTT medium at 32 °C until exponential phase and harvested by centrifugation for 10 min at 4150 × g. The cell pellet was resuspended in lysis buffer (6.7 mM 2-(N-morpholino)ethanesulfonic acid (MES), 6.7 mM (4-(2-hydroxyethyl)-1-piperazineethanesulfonic acid (HEPES), 200 mM NaCl, 6.7 mM potassium acetate (KAc), pH 7.5) containing cOmplete™, Mini Protease Inhibitor Cocktail (Roche), resulting in 250 times concentrated cells. Cells were lysed three times through a French Press and then centrifuged at 109,760 × g rpm for 1 h at 4 °C. The protein concentration of the supernatant was determined using a Bradford protein assay and 233 μg protein were mixed with 20 μl 5× capture buffer (100 mM HEPES, 250 mM KAc, 50 mM magnesium acetate (MgAc), 50% glycerol, pH 7.5) and 10 μM c-di-GMP capture compound. The reaction volume was adjusted with H₂O to 100 μl. In the negative control, c-di-GMP capture compound was replaced by 1× capture buffer. In the second competition control, 1 mM c-di-GMP (BioLog) was added and incubated for 30 min prior to capture compound addition. Reactions were incubated for 2 h at 4 °C in the dark on a rotary wheel. After UV irradiation for 4 min at 4 °C in a caproBox, 50 μl magnetic streptavidin beads (Thermo Fisher Scientific) and 25 μl 5× wash buffer (250 mM Tris HCl pH 7.5, 5 M NaCl, 0.1% n-octyl-β-glucopyranoside) were added to the reaction and the mixture was incubated for 1 h at 4 °C

on a rotary wheel. The beads were then collected with a magnet and washed six times with 200 μl 1× wash buffer, one time with 200 μl H₂O and six times with 80% acetonitrile and resuspended in 200 μl H₂O. 7 μl 1 M ammonium bicarbonate, 17 μl 100 mM DTT and 5.1 μl 0.1 μg μl⁻¹ Trypsin (Promega) were added to the beads and incubated for 12 h at 25 °C. Beads were removed with a magnet, 3 μl of AcOH was added and samples dried in a speedvac for 50 min. Captured proteins were identified by MALDI-TOF-TOF mass-spectrometry analysis as described[83].

**Preparation of [α-³²P]-labeled c-di-GMP**. [α-³²P]-labeled c-di-GMP was prepared as described previously[26] by incubating 10 μM His₆-DgcA (final concentration) with 1 mM GTP/[α-³²P]-GTP (0.1 μCi μl⁻¹) in reaction buffer (50 mM Tris-HCl pH 8.0, 300 mM NaCl, 10 mM MgCl₂) in a total volume of 200 μl overnight at 30 °C. The reaction mixture was then incubated with five units of calf intestine alkaline phosphatase (Fermentas) for 1 h at 22 °C to hydrolyze unreacted GTP. The reaction was stopped by incubation for 10 min at 95 °C. The reaction was centrifuged (10 min, 15,000 × g, 20 °C) and the supernatant was used for the c-di-GMP binding assay.

**In vitro c-di-GMP binding assays**. In the DRaCALA assay[36,84] [α-³²P]-c-di-GMP was mixed with 20 μM of the relevant protein and incubated for 10 min at 30 °C in binding buffer (10 mM TrisHCl pH 8.0, 100 mM NaCl, 5 mM MgCl₂). Ten microliter of this mixture was transferred to a nitrocellulose filter (GE Healthcare), allowed to dry and imaged using a STORM 840 Scanner (Amersham Biosciences). For competition experiments, 0.4 mM unlabeled c-di-GMP (BioLog) or GTP (Sigma-Aldrich) was used. ITC was performed with the ITC₂₀₀-System (MicroCal) at 25 °C. CdbA (200 μl, 25 μM) was titrated with 2 μl per injection of 150 μM c-di-GMP for a total of 19 injections. Every injection was applied over a period of 4 s and individual injections were separated by breaks of 180 s. To subtract dilution heat of c-di-GMP, a measurement with the same protocol was performed using only SEC buffer in the sample cell. Data analysis was done with NITPIC 1.2.2[85] and the experimental data fitted using SEDPHAT 12.1b[86]. The final image was generated with GUSSI 1.3.2[87].

**Hydrogen-deuterium exchange mass-spectrometry (HDX-MS)**. Sample preparation for HDX-MS was aided by a two-arm robotic autosampler (LEAP Technologies). CdbA (7.5 μl, 50 μM) with or without 500 μM c-di-GMP (BioLog) was mixed with 67.5 μl D₂O-containing SEC buffer (20 mM of HEPES-Na, pH 7.5, 200 mM NaCl 20 mM MgCl₂, 20 mM KCl). After incubation for 10/30/95/1000/10000 s at 25 °C, 55 μl of the HDX reaction were added to 55 μl quench solution (400 mM KH₂PO₄/H₃PO₄, 2 M guanidine-HCl, pH 2.2) kept at 1 °C and 95 μl of the mixture injected into an ACQUITY UPLC M-class system with HDX technology (Waters)[88]. Peptides were generated online using immobilized pepsin at 12 °C and 100 μl/min flow rate of water + 0.1% (v/v) formic acid and the resulting peptic peptides trapped on a C18 column (Waters) kept at 0.5 °C. After 3 min, the trap column was placed in line with an ACQUITY UPLC BEH C18 1.7 μm 1.0 × 100 mm column (Waters) and the peptides eluted at 0.5 °C using a gradient of water + 0.1% (v/v) formic acid (eluent A) and acetonitrile + 0.1% (v/v) formic acid (eluent B) at 30 μl/min flow rate: 0-7 min/95-65% A, 7-8 min/65-15% A, 8–10 min/15% A, 10-11 min/5% A, 11-16 min/95% A. Undeuterated samples were obtained by similar procedure through dilution in H₂O-containing SEC buffer. Mass spectra were recorded in positive ion mode using a Synapt G2-Si HDMS mass spectrometer equipped with an ESI source (Waters) in HDMS (High Definition-MS) or HDMS^E (Enhanced High Definition MS)[89,90] mode for deuterated and undeuterated samples, respectively. [Glu1]-Fibrinopeptide B standard (Waters) was used for lock mass correction. The pepsin column was washed three times with 80 μl of 4% (v/v) acetonitrile and 0.5 M guanidine-HCl during each run and additionally blank runs were performed between each sample to avoid peptide carry-over. All measurements were performed in triplicates. Peptide identification and assignment of deuterium incorporation was carried out as described previously[91–93] aided by PLGS and DynamX 3.0 softwares (Waters). A total of 68 peptides were obtained that covered the entire sequence of CdbA with 11-fold redundancy per amino acid. HDX-MS data can be found in the Supplementary Data 1[94,95].

**Chromatin Immunoprecipitation Sequencing (ChIP-seq)**. Chromatin immuno-precipitation was performed essentially as described before[96]. Briefly, cultures (50 ml) of each ChIP strain (ParB-3×FLAG, CdbA_3×FLAG, WT) were grown to mid-late exponential phase (OD₆₀₀ = 0.5–0.9) at 32 °C with aeration. Crosslinking was initiated by the addition of 1% formaldehyde in the presence of 10 mM sodium phosphate (pH 7.6) for 10 min at room temperature with shaking (100 rpm). After cooling in ice for 30 min, cells were quenched with 125 mM glycine, pelleted, washed twice with phosphate-buffered saline solution and stored at −80 °C until further use. The frozen pellet was thawed and resuspended and lysed in TES buffer (470 μl, 10 mM Tris-HCl pH 7.6, 1 mM EDTA, 100 mM NaCl, 2.2 mg ml⁻¹ lysozyme) in the presence of 1× complete protease inhibitor cocktail (Roche) for 30 min at 37 °C with shaking (100 rpm). It was mixed with ChIP buffer (550 μl, 1.1% Triton X-100, 1.2 mM EDTA, 16.7 mM Tris pH 8.1, 167 mM NaCl, 1× complete protease inhibitor cocktail), incubated for 10 min at 37 °C and sonicated to generate fragments of ~200-400 bp. The sample was clarified by centrifugation and an aliquot of 50 μl supernatant was kept as the input sample. The rest was mixed with

550 µl ChIP buffer with 0.01% sodium dodecyl sulfate (SDS) and pre-cleaned with 30 µl of protein A magnetic beads (Thermo Scientific™) previously washed with PBS buffer (37 mM NaCl, 2.7 mM KCl, 10 mM $Na_2HPO_4$, 2 mM $K_2HPO_4$) plus 1 mgml$^{-1}$ bovine serum albumin (BSA). 10 µl of monoclonal α-FLAG (Rockland) antibodies was added, incubated at 4 °C overnight with rotation and immuno-precipitated (2.5 h, 4 °C) with rotation with 30 µl of protein A magnetic Dynabeads previously washed with PBS plus 1 mg ml$^{-1}$ BSA. The beads were washed with low salt buffer (0.1% SDS, 1% Triton X-100, 2 mM EDTA, 20 mM Tris pH 8.1, 150 mM NaCl), with high salt buffer (0.1% SDS, 1% Triton X-100, 2 mM EDTA, 20 mM Tris pH 8.1, 0.5 M NaCl), with LiCl buffer (0.25 M LiCl, 1% NP-40, 1% sodium deoxycholate, 1 mM EDTA, 10 mM Tris pH 8.1), and twice with TE buffer (10 mM Tris pH 8.0, 1 mM EDTA). The protein–DNA complex eluted from the beads in two 100 µl fractions of elution buffer (1% SDS, 0.1 M NaHCO$_3$) and incubated at 65 °C overnight in the presence of 20 µl 3 M NaCl to disrupt the crosslinks, treated with 0.5 µl proteinase K (50 mg ml$^{-1}$) at 42 °C for 2 h and with 2 µl RNase A (10 mg ml$^{-1}$) at 37 °C for 20 min, followed by DNA isolation using PCR product purification kit (Macherey-Nagel). The input sample was also subjected to this cross-link reversal and DNA extraction protocol. FS libraries were prepared and sequenced (paired-end, 2×150 bp) on an Illumina HiSeq3000 instrument at the Max Planck-Genome-centre Cologne.

**ChIP-seq data analysis.** CLC workbench 12.0 (Qiagen, Hilden, Germany) was used for computational processing of sequencing data. The functions 'Trim Reads', 'Map Reads to Reference', 'Duplicate Mapped Reads Removal', 'Transcription Factor ChIP-Seq' & 'Annotate with Nearby Gene Information' were applied with default settings. ChIP-Seq peak calling in CLC workbench is based on a shape learning algorithm[97]. Here every two samples were compared individually to their respective controls where the genome assembly of *Myxococcus xanthus* DK1622 (Accession: NC_008095) obtained from NCBI was used as reference. Sequence coverage of peaks was inferred via samtools[98] (option 'mpileup', Version 1.9). Peaks were considered significant if the enrichment in the sample over input was ≥4-fold. The DNA binding consensus of CdbA was identified using sequences from the top 100 or top 500 peaks (peak summit ±50 bp) and the MEME-ChIP web tool[99]. The probability matrix from the MEME-ChIP analysis using the top 100 peaks served as an input to create the Hidden Markov Model used for the genome-wide search for putative CdbA binding sites with a given score. P-value of identified sequences was determined based on in silico simulation. In this simulation 1,000,000 of 18 bp random sequences with the same GC content as *M. xanthus* genome were screened for the presence of the same probability matrix with a given score. A motif was assigned to the peak if it was found within a range of a peak summit ±50 bp.

**Bioinformatics.** Homology based structure prediction for CdbA and CdbB was performed using HHPred[100]. 16 s rRNA and protein sequences were aligned with ClustalW using MEGA7[101] and the 16 s rRNA phylogenetic tree was generated using the Maximum Likelihood method. CdbA/B homologs were identified using BLASTP[102] analysis and the sequence identity/similarity was calculated using EMBOSS Needle software[103] (pairwise sequence alignment).

**Statistics.** All statistics were performed using a two-tailed Student's *t*-test for samples with unequal variances using Sigmaplot 12.5 and Microsoft Excel 2013.

**Plasmid construction.** Primers used in this study are listed in Supplementary Table 4.

pDJS 83 (for overexpression and purification of CdbB), *cdbB* was amplified from DK1622 genomic DNA with the primer pair "4362 F"/"4362 –stop R", digested with NdeI and HindIII, cloned into pET24b+ digested with the same enzymes and sequenced.

pDJS 85 (for generation of in-frame deletion of *cdbA-B*), up- and downstream fragments were amplified from DK1622 genomic DNA using the primer pairs "4361-2_A"/"4361-2_B (5aa)" and "4361-2_C (5aa)"/"4361-2_D", as described[104]. The AB and CD fragments were used for overlapping PCR with the primer pair "4361-2_A"/"4361-2_D" to generate the AD fragment. The AD fragment was digested with KpnI and XbaI, cloned into pBJ114 digested with the same enzymes and sequenced.

pDJS 86 (for overexpression and purification of CdbA), *cdbA* was amplified from DK1622 genomic DNA with the primer pair "4361 Fw"/"4361 -stop Rev", digested with NdeI and HindIII, cloned into pET24b+ digested with the same enzymes and sequenced.

pDJS 97 (for generation of in-frame deletion of *cdbB*), up- and downstream fragments were amplified from DK1622 genomic DNA using the primer pairs "4362_A"/"4362_B 5aa" and "4362_C 5aa"/"4361-2_D", as described[104]. The AB and CD fragments were used for overlapping PCR with the primer pair "4362_A"/"4361-2_D" to generate the AD fragment. The AD fragment was digested with KpnI and XbaI, cloned into pBJ114 digested with the same enzymes and sequenced.

pDJS 99 (for generation of in-frame deletion of *cdbA*), up- and downstream fragments were amplified from DK1622 genomic DNA using the primer pairs "4361-2_A"/"4361_B 5aa" and "4361_C 5aa"/"4361_D 5aa" as described[104]. The AB and CD

fragments were used for overlapping PCR with the primer pair "4361-2_A"/"4361_D 5aa" to generate the AD fragment. The AD fragment was digested with KpnI and XbaI, cloned into pBJ114 digested with the same enzymes and sequenced.

pDJS 105 (for overexpression and purification of AmrZ), *amrZ* was amplified from *P. aeruginosa* PA01 genomic DNA with the primer pair "amrZ F"/"amrZ -stop R", digested with NdeI and HindIII, cloned into pET24b+ digested with the same enzymes and sequenced.

pDJS 107 (for overexpression and purification of CdbA$^{K8A/S10A}$), *cdbA*$^{K8A/S10A}$ was amplified from DK1622 genomic DNA with the primer pair "4361 AQA Fw"/"4361 -stop Rev", digested with NdeI and HindIII, cloned into pET24b+ digested with the same enzymes and sequenced.

pDJS 125 (for expression of Pnat_*cdbA*_mCherry from the *attB* site), the Pnat_*cdbA* was amplified from DK1622 genomic DNA with the primer pair "Pnat 4361 F KpnI"/"4361 -st R BamHI", digested with KpnI and BamHI, cloned into pNG62 digested with the same enzymes and sequenced. pNG62 is a derivative of pSWU19 (multi-cloning-site, linker, mCherry) and was a gift from N. Gomez-Santos.

pDJS 127 (for overexpression and purification of CdbA$^{R27A/R30A}$), *cdbA*$^{R27A/R30A}$ was amplified from DK1622 genomic DNA with the primer pairs "4361 Fw"/"4361 R27,30 A R" and "4361 R27,30 A F"/"4361 -stop Rev". Resulting fragments were used as a template for overlapping PCR with the primer pair "4361 Fw"/"4361 -stop Rev". Resulting fragment was digested with NdeI and HindIII, cloned into pET24b+ digested with the same enzymes and sequenced.

pDJS 129 (for expression of Pnat_*cdbA*$^{K8A/S10A}$ from the *attB* site), Pnat_*cdbA*$^{K8A/S10A}$ was amplified from DK1622 genomic DNA with the primer pairs "Pnat 4361 F EcoRI"/"4361 AQA -" and "4361 AQA +"/"4361 Rev". Resulting fragments were used as a template for overlapping PCR with the primer pair "Pnat 4361 F EcoRI"/"4361 Rev". Resulting fragment was digested with EcoRI and HindIII, cloned into pSW105 digested with the same enzymes and sequenced.

pDJS 130 (for expression of Pnat_*cdbA*$^{R27A/R30A}$ from the *attB* site), Pnat_*cdbA*$^{R27A/R30A}$ was amplified from DK1622 genomic DNA with the primer pairs "Pnat 4361 F EcoRI"/"4361 R27,30 A R" and "4361 R27,30 A F"/"4361 Rev". Resulting fragments were used as a template for overlapping PCR with the primer pair "Pnat 4361 F EcoRI"/"4361 Rev". Resulting fragment was digested with EcoRI and HindIII, cloned into pSW105 digested with the same enzymes and sequenced.

pDJS 132 (BTH plasmid for *cdbA*), *cdbA* was amplified from DK1622 genomic DNA with the primer pairs "BTH 4361 XbaI fw pKT25/pUT18C"/"BTH 4361 KpnI rev stop pKT25/pUT18C".Resulting fragment was digested with XbaI and KpnI, cloned into pKT25 digested with the same enzymes and sequenced.

pDJS 133 (BTH plasmid for *cdbA*), *cdbA* was amplified from DK1622 genomic DNA with the primer pairs "BTH 4361 XbaI fw pKNT25/pUT18"/"BTH 4361 KpnI rev pKNT25/pUT18". Resulting fragment was digested with XbaI and KpnI, cloned into pNKT25 digested with the same enzymes and sequenced.

pDJS 134 (BTH plasmid for *cdbA*), *cdbA* was amplified from DK1622 genomic DNA with the primer pairs "BTH 4361 XbaI fw pKNT25/pUT18"/"BTH 4361 KpnI rev pKNT25/pUT18". Resulting fragment was digested with XbaI and KpnI, cloned into pUT18 digested with the same enzymes and sequenced.

pDJS 135 (BTH plasmid for *cdbA*), *cdbA* was amplified from DK1622 genomic DNA with the primer pairs "BTH 4361 XbaI fw pKT25/pUT18C"/"BTH 4361 KpnI rev stop pKT25/pUT18C". Resulting fragment was digested with XbaI and KpnI, cloned into pUT18C digested with the same enzymes and sequenced.

pDJS 136 (BTH plasmid for *cdbB*), *cdbB* was amplified from DK1622 genomic DNA with the primer pairs "BTH 4362 XbaI fw pKT25/pUT18C"/"BTH 4362 KpnI rev stop pKT25/pUT18C". Resulting fragment was digested with XbaI and KpnI, cloned into pKT25 digested with the same enzymes and sequenced.

pDJS 137 (BTH plasmid for *cdbB*), *cdbB* was amplified from DK1622 genomic DNA with the primer pairs "BTH 4362 XbaI fw pKNT25/pUT18"/"BTH 4362 KpnI rev pKNT25/pUT18". Resulting fragment was digested with XbaI and KpnI, cloned into pNKT25 digested with the same enzymes and sequenced.

pDJS 138 (BTH plasmid for *cdbB*), *cdbB* was amplified from DK1622 genomic DNA with the primer pairs "BTH 4362 XbaI fw pKNT25/pUT18"/"BTH 4362 KpnI rev pKNT25/pUT18". Resulting fragment was digested with XbaI and KpnI, cloned into pUT18 digested with the same enzymes and sequenced.

pDJS 139 (BTH plasmid for *cdbB*), *cdbB* was amplified from DK1622 genomic DNA with the primer pairs "BTH 4362 XbaI fw pKT25/pUT18C"/"BTH 4362 KpnI rev stop pKT25/pUT18C". Resulting fragment was digested with XbaI and KpnI, cloned into pUT18C digested with the same enzymes and sequenced.

pDJS 143 (for expression of *cdbA*-mCherry in *E.coli*), the *cdbA*-mCherry was amplified from pDJS125 with the primer pair "4361 Fw"/"DSZ 24", digested with NdeI and KpnI, cloned into pRSFDuet-1 digested with the same enzymes and sequenced.

pDJS 148 (for expression of *cdbA*-mCherry from vanilate promoter from mx18-19 site), the *cdbA*-mCherry fragment was amplified from pDJS125 with the primer pair "4361-2 Fw"/"GFP rv_STOP_EcoRI", digested with NdeI and EcoRI, cloned into pMR3691 digested with the same enzymes and sequenced.

pDJS 151 (for expression of Pnat_*parB*_YFP from the *attB* site), Pnat_*parB*_YFP was amplified from pAH07 with the primer pair "DA-325"/"KA439", digested with NdeI and XbaI, cloned into pSWU19 digested with the same enzymes and sequenced.

pDJS 154 (for expression of Pnat_*cdbA*$^{K8A/S10A}$_mCherry from the *attB* site), Pnat_*cdbA*$^{K8A/S10A}$ was amplified from pDJS129 with the primer pair "Pnat 4361 F

KpnI"/"4361 -st R BamHI", digested with KpnI and BamHI, cloned into pNG62 digested with the same enzymes and sequenced.

pDJS 155 (for expression of Pnat_$cdbA$^R27A/R30A_mCherry from the $attB$ site), Pnat_$cdbA$^R27A/R30A fragment was amplified from pDJS130 with the primer pair "Pnat 4361 F KpnI"/"4361 -st R BamHI", digested with KpnI and BamHI, cloned into pNG62 digested with the same enzymes and sequenced.

pDJS 170 (for replacement of $cdbA$ with $cdbA$-3xFLAG at native site), up- and downstream fragments were amplified from DK1622 genomic DNA using the primer pairs "4361-2_A"/" 4361 3xFLAG rev" and "4361 3xFLAG fw"/"4361-2_D". Resulting fragments were used as a template for overlapping PCR with the primer pair "4361-2_A"/"4361-2_D". Resulting fragment was digested with KpnI and XbaI, cloned into pBJ114 digested with the same enzymes and sequenced.

pDJS 177 (for overexpression and purification of His₆_CdbA_3xFLAG), the $cdbA$_3xFLAG fragment was amplified from pDJS 170 with the primer pair "4361 no start NdeI"/"4361 3xFLAG stop BamHI", digested with NdeI and BamHI, cloned into pET28a+ digested with the same enzymes and sequenced.

pDJS 179 (for replacement of $parB$ with $parB$-3xFLAG at native site), up- and downstream fragments were amplified from DK1622 genomic DNA using the primer pairs "ParB 3xFLAG A"/"ParB 3xFLAG B" and "ParB 3xFLAG C"/"ParB 3xFLAG D". Resulting fragments were used as a template for overlapping PCR with the primer pair "ParB 3xFLAG A"/"ParB 3xFLAG D". Resulting fragment was digested with EcoRI and XbaI, cloned into pBJ114 digested with the same enzymes and sequenced.

**Reporting summary**. Further information on research design is available in the Nature Research Reporting Summary linked to this article.

## Data availability

Co-ordinates and structure factors have been deposited with the RCSB: accession codes of 6SBW (form one) and 6SBX (form two). The ArrayExpress accession number for the ChIP-seq experiment is E-MTAB-8535. The HDX-MS data have been deposited to the ProteomeXchange Consortium via the PRIDE partner repository with the dataset identifier PXD018028. The source data underlying Figs. 1b, d, 4a, c–e, 6a, c, d, 7a–e, 8a, c and Supplementary Figs. 3b–d, 4a, d, e, 5, 6b, 7a, 8a, b are provided as a Source Data file.

## Code availability

Custom MATLAB scripts used for ChIP-seq data analysis and for microscopy analysis are available from the corresponding author upon request.

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

## Acknowledgements

We thank Jörg Kahnt for mass-spectrometry analysis, Dr. Nuria Gómez Santos for providing the plasmid pNG62, Prof. Dr. Anke Becker for helpful discussions and Max Planck Genome Centre Cologne (https://mpgc.mpipz.mpg.de/home) for ChIP-seq library construction and next-generation sequencing. This work was supported by funding from the Deutsche Forschungsgemeinschaft (DFG) within the framework of the Collaborative Research Center SFB987 "Microbial Diversity in Environmental Signal Response" (to L.S.-A.) and the priority programme SPP 1879 "Nucleotide Second Messenger Signaling in Bacteria" (to L.S.-A. and G.B.) and the Max Planck Society (to L.S.-A.). We acknowledge support by the Deutsche Forschungsgemeinschaft (DFG) through the DFG core facility for interactions, dynamics, and macromolecular assembly structure (to G.B.).

## Author contributions

D.S. and L.S.-A. conceptualized the study. D.S., W.S., and I.T.C. conducted the investigation. D.S. and D.Sz. developed the methodology, analyzed microscopy images, and ChIP-seq data. D.S. and L.S.-A. wrote the original draft of the manuscript. G.B., A.L.L., W.S., D.Sz., and I.T.C. reviewed and edited the manuscript. G.B. and L.S.-A. acquired funding. G.B., A.L.L., and L.S.-A. provided supervision

## Competing interests

The authors declare no competing interests.
