## [Peer Review File · Nature Communications]

Reviewer #1 (Remarks to the Author):

This paper describes the characterisation of a small (67 aa) DNA-binding protein in *M. xanthus* the authors name CdbA, and its homolog, CdbB. The authors identify these 2 proteins as c-di-GMP-binding proteins using 'capture compound' technology. They show that CdbA is essential, they determine the K_d of the protein for c-di-GMP (~83 nM) and they determine the crystal structure of CdbA (in the absence of c-di-GMP). Using ChIP-seq they show that CdbA binds to 100s of sites across the chromosome, most of them within open reading frames (~35% mapping to intergenic regions). Engineered depletion of the essential CdbA protein has striking phenotypes, including defects in chromosome organisation and segregation, resulting in filamentation and death. Using in vitro DNA-binding assays, the authors show that c-di-GMP blocks DNA binding by CdbA and they present evidence that DNA binding and c-di-GMP binding are mutually exclusive. They conclude that CdbA is a NAP (not a transcription factor) involved in nucleoid organisation, and that its activity is modulated by c-di-GMP. Paradoxically, however, the authors do not see or any effects on chromosome organisation (ie any cdbA mutant-like phenotypes) when they engineer artificially high (or low) c-di-GMP levels in vivo.

This manuscript represents a great deal of hard work, the experiments are well performed, the paper is well written, and the story is very interesting. Overall I think it merits publication in Nature Communications.

1. I believe the authors' data (ie I believe that cdbA is essential, that CdbA is NAP, and that c-di-GMP can prevent CdbA binding to DNA). The paradox then, is why engineering high levels of c-di-GMP has no phenotypic consequences with regard to chromosome organisation and segregation, when high c-di-GMP in principle might be expected to mimic the very striking phenotypes seen when the essential protein CdbA is depleted from the cell. The authors speculate in the Discussion that c-di-GMP may 'fine tune' DNA binding by CdbA, rather than elicit an all-or-none response, but it is still hard to rationalise why engineering high c-di-GMP levels has no phenotypic consequences, given the striking cdbA null phenotype. Beyond the scope of the current study, it would be interesting to perform ChIP-seq in the strains engineered to have high or low c-di-GMP, in parallel to the WT, to see if any differences in the global pattern of CdbA binding across the chromosome can be detected.
2. Further, given that c-di-GMP levels rise steeply during development in *M. xanthus*, in the future it would be very interesting to do time-resolved ChIP-seq across the life cycle.
3. Related to (2), The measured in vivo [c-di-GMP] concentration during growth in rich medium is ~1.4 μM, implying (naïvely at least) that CdbA would be near-saturated with c-di-GMP in vivo and therefore not bound to DNA, which is clearly not true based on eg ChIP-seq and the cell biology.
4. A further interesting observation is that the K_d for c-di-GMP binding to free CdbA is ~83 nM as measured by ITC, but the authors had to use very much higher concentrations of c-di-GMP to prevent DNA binding by CdbA in EMSA assays. The authors speculate that the K_d for c-di-GMP in the DNA-bound form may be much higher. This could be tested by order-of-addition EMSA experiments, adding c-di-GMP to CdbA first then DNA second, and vice versa, with DNA added to CdbA first then c-di-GMP second. Does the order of addition make a difference? In which order were the c-di-GMP and DNA added in the experiments already presented in the paper (state this in the MS)?
5. I found it odd that the authors made RNA from the WT and the CdbA-depleted strain, but then used it as a template for qRT-PCR of a few chosen CdbA 'target genes', rather than subjecting the samples to deep sequencing to gain a much more informative chromosome-wide picture of the effects of CdbA depletion on global gene expression. Why? In our experience, it's actually more work to do the qRT-PCR than to do the deep sequencing, plus it's less informative!

6. I think the distinction between a 'NAP' and a 'transcription factor' is not quite as clear cut as depicted in the MS and could be seen as more of a continuum. For example, FIS does have clear and quite strong regulatory influence at some of its binding sites. Ironically, one of the most commonly stated distinctions between NAPs and transcription factors is that the activities of NAPs are often said to be determined by abundance alone and are not subject to post-translational regulation by e.g. small molecules. The control of CdbA by c-di-GMP makes it a 'ligand-regulated NAP', further blurring these definitions.

Reviewer #2 (Remarks to the Author):

The manuscript by Skotnicka et al describes the identification, and biochemical and biophysical characterization of the CdbA protein and how it binds DNA and the molecule di-c-GMP. Using a combination of X-ray crystallography, HDX-MS and cellular assays they defined the molecular basis for its regulation by di-c-GMP and its role in promoting cellular growth. They identified multiple mutations that fully disrupt DNA and di-c-GMP binding. They have mapped potential DNA binding sites, and showed a clear role in cellular growth.

Overall, this is an interesting manuscript, which provides unique molecular insight into the role of a novel DNA binding protein, that can be regulated by the messenger di-c-GMP. I am quite positive in my assessment, although certain aspects of the presentation could be improved to increase the clarity and accessibility to a broad audience.

Major points

1. The HDX-MS experiments are well described in the methods. However how it is presented could be improved. Figure 2 that shows the HDX data and crystallographic data is very hard to interpret. First in Fig. 2b the deuterium incorporation is shown at a per residue basis, which is calculated from overlapping peptides. The data was performed in triplicate as described in methods, but no data is provided whatsoever on the error within the experiments. This needs to be incorporated into the figures/supplement.
2. Ideally the full HDX dataset should be included in excel format as suggested by the recent community guidelines as published in Nature Methods (Masson et al 2019). With these details included it would be much easier to interpret the data.
3. It is also not entirely clear how the structures are colored for the HDX-MS analysis of the di-c-GMP interface as all residues from 1-30 are protected, yet only a selection of those residues is mapped on the structure and described in text. The region from 12-20 is colored in Fig. 2b, yet is not described. Is this change really significant in these regions? More details on the raw deuterium incorporation and the error of the experiments would make this much clearer.
4. The overlay of the CdbA protein and its putative DNA binding interface compared to Arc is very difficult to interpret as currently presented in Fig. 2e. It would be useful to also include an electrostatic surface of CdbA to examine the feasibility of the putative DNA binding surface.
5. The mutants generated that disrupt DNA and di-c-GMP binding are a very nice addition to the manuscript. It is not entirely clear how different the mutants are to WT, it might be useful to compare on the same gel filtration trace the mutant vs wild type. It almost appears that the mutants are more closely mimicking the WT bound to di-c-GMP state (even though they cannot bind this molecule).

Reviewer #3 (Remarks to the Author):

In this manuscript by Skotnicka et al, the authors identified CdbAB as c-di-GMP binding proteins, solved the crystal structure of CdbA, identified residues in CdbA that were important for c-di-GMP binding, and found that mutants that did not bind c-di-GMP abolished DNA binding. Using ChIP-seq and fluorescence microscopy, the authors found that CdbA bound to the chromosome globally, and that depletion of CdbA caused defects in chromosome segregation and organization, and ultimately led to cell division defects. Because CdbA had little effect on transcription, the authors concluded that CdbA is a nucleoid associated protein, whose activity could be modulated by c-di-GMP. This is the first example that the second messenger c-di-GMP could play a role in chromosome organization and segregation. However, the authors also showed that modulating c-di-GMP levels in vivo by expressing heterologous DGC or PDE had no effect on cell growth or chromosome biology. Therefore, perhaps beyond the scope of this study, whether and how c-di-GMP modulates CdbA activity in vivo are still to be addressed.

This is a solid study involving a comprehensive set of approaches and assays. The experiments were carefully executed, the results were critically analyzed and interpreted, and the conclusions were well supported by experimental data.

Minor points:

In Figure 6f, DAPI-stained nucleoid should be colocalized in the same cells to show whether the CdbA*-mCherry colocalizes with the nucleoid, to support conclusions of Line 339-341: "The two variants also failed to localize over the nucleoid in *M. xanthus* (Fig. 6f), confirming the in vitro result that they have a DNA binding defect (Fig. 6c)."

Reviewers' comments:

Reviewer #1 (Remarks to the Author):

This paper describes the characterisation of a small (67 aa) DNA-binding protein in *M. xanthus* the authors name CdbA, and its homolog, CdbB. The authors identify these 2 proteins as c-di-GMP-binding proteins using 'capture compound' technology. They show that CdbA is essential, they determine the K_d of the protein for c-di-GMP (~83 nM) and they determine the crystal structure of CdbA (in the absence of c-di-GMP). Using ChIP-seq they show that CdbA binds to 100s of sites across the chromosome, most of them within open reading frames (~35% mapping to intergenic regions). Engineered depletion of the essential CdbA protein has striking phenotypes, including defects in chromosome organisation and segregation, resulting in filamentation and death. Using in vitro DNA-binding assays, the authors show that c-di-GMP blocks DNA binding by CdbA and they present evidence that DNA binding and c-di-GMP binding are mutually exclusive. They conclude that CdbA is a NAP (not a transcription factor) involved in nucleoid organisation, and that its activity is modulated by c-di-GMP. Paradoxically, however, the authors do not see or any effects on chromosome organisation (ie any cdbA mutant-like phenotypes) when they engineer artificially high (or low) c-di-GMP levels in vivo.

This manuscript represents a great deal of hard work, the experiments are well performed, the paper is well written, and the story is very interesting. Overall I think it merits publication in Nature Communications.

Response: Thank you very much!

1. I believe the authors' data (ie I believe that cdbA is essential, that CdbA is NAP, and that c-di-GMP can prevent CdbA binding to DNA). The paradox then, is why engineering high levels of c-di-GMP has no phenotypic consequences with regard to chromosome organisation and segregation, when high c-di-GMP in principle might be expected to mimic the very striking phenotypes seen when the essential protein CdbA is depleted from the cell. The authors speculate in the Discussion that c-di-GMP may 'fine tune' DNA binding by CdbA, rather than elicit an all-or-none response, but it is still hard to rationalise why engineering high c-di-GMP levels has no phenotypic consequences, given the striking cdbA null phenotype. Beyond the scope of the current study, it would be interesting to perform ChIP-seq in the strains engineered to have high or low c-di-GMP, in parallel to the WT, to see if any differences in the global pattern of CdbA binding across the chromosome can be detected.

Response: We agree with the reviewer that the lack of phenotypic consequences of increased c-di-GMP level is interesting and paradoxical. In the Discussion we offer two explanations (line 498-503) (1) "...alteration in c-di-GMP may not elicit an all-or-none response with respect to CdbA DNA binding but rather a graded response in which c-di-GMP binding to CdbA modulates or fine-tunes CdbA DNA binding in vivo"; and (2) "Alternatively, exposure of CdbA to c-di-GMP is controlled by complex formation with specific DGCs/PDEs creating a local c-di-GMP pool. In this scenario, manipulating the global c-di-GMP level would not affect CdbA-DNA binding". It is also possible that we are not reaching a sufficiently high c-di-GMP level in this strain to cause a response. It is a very good idea to perform a ChIP-seq on those strains and we are definitely going to do that in the future.

2. Further, given that c-di-GMP levels rise steeply during development in *M. xanthus*, in the future it would be very interesting to do time-resolved ChIP-seq across the life cycle.

Response: Thank you; this is a great idea and we are definitely also going to do that in the future.

3. Related to (2), The measured in vivo [c-di-GMP] concentration during growth in rich medium is ~1.4 μ M, implying (naïvely at least) that CdbA would be near-saturated with c-di-GMP in vivo and therefore not bound to DNA, which is clearly not true based on eg ChIP-seq and the cell biology.

Response: The reviewer raises an interesting point. One explanation could be the high abundance of CdbA in the cell. From quantitative immunoblots, we estimate that in vivo the CdbA tetramer concentration is ~2.2 μM (line 302). One tetramer can bind 2 molecules of c-di-GMP. Given that the in vivo concentration of c-di-GMP is ~1.4 μM , a fraction of CdbA will likely be in the non-c-di-GMP bound form, which can bind DNA.

4. A further interesting observation is that the K_d for c-di-GMP binding to free CdbA is ~83 μM as measured by ITC, but the authors had to use very much higher concentrations of c-di-GMP to prevent DNA binding by CdbA in EMSA assays. The authors speculate that the K_d for c-di-GMP in the DNA-bound form may be much higher. This could be tested by order-of-addition EMSA experiments, adding c-di-GMP to CdbA first then DNA second, and vice versa, with DNA added to CdbA first then c-d-GMP second. Does the order of addition make a difference? In which order were the c-di-GMP and DNA added in the experiments already presented in the paper (state this in the MS)?

Response: That is also a very interesting point. In the experiment in Fig. 7a, we first added different concentrations of c-di-GMP to CdbA for 10 min and then the DNA probe (this is now specifically stated in figure legend). Inspired by the reviewer's comment we have included a new "order of addition experiment" in Supplementary Fig. 5 and described in line 336-340. We did not observe different effects of c-di-GMP on CdbA DNA binding in these experiments. Therefore, we speculate in the Discussion (line 490-493) that "...DNA-binding changes CdbA structure/dynamics such that c-di-GMP binding is less favorable than when DNA is absent (i.e. this is not represented by a simple competition model, and can be enacted by an extensive interface for DNA binding that is more than just the two β -sheets)".

5. I found it odd that the authors made RNA from the WT and the CdbA-depleted strain, but then used it as a template for qRT-PCR of a few chosen CdbA 'target genes', rather than subjecting the samples to deep sequencing to gain a much more informative chromosome-wide picture of the effects of CdbA depletion on global gene expression. Why? In our experience, it's actually more work to do the qRT-PCR than to do the deep sequencing, plus it's less informative!

Response: We agree with the reviewer that, in principle, RNA-seq experiments can be more informative than qRT-PCR experiments. However, we decided for the qRT-PCR analysis of specific genes because we isolate total RNA from cells that we are depleting of CdbA. Because CdbA depletion is lethal, we were concerned that we would observe many changes in gene expression using an RNA-seq approach that would not be directly related to CdbA depletion but only indirectly related and caused by cells undergoing cell death. Hence, we focused our analysis on a set of carefully selected genes.

6. I think the distinction between a 'NAP' and a 'transcription factor' is not quite as clear cut as depicted in the MS and could be seen as more of a continuum. For example, FIS does have clear and quite strong regulatory influence at some of its binding sites. Ironically, one of the most commonly stated distinctions between NAPs and transcription factors is that the activities of NAPs are often said to be determined by abundance alone and are not subject to post-translational regulation by e.g. small molecules. The control of CdbA by c-di-GMP makes it a 'ligand-regulated NAP', further blurring these definitions.

Response: We agree with the reviewer that the distinction between NAPs and transcription factors is getting increasingly blurred (and our findings that CdbA is a ligand-regulated NAP, contributes further to this blurring). However, our classification of CdbA as a NAP is based on several features that CdbA has in common with classical, well-defined NAPs (i.e. binding DNA with moderate sequence specificity causing bending, wrapping or bridging of DNA and, typically, only regulating transcription on a small scale). Therefore, we would like to avoid including a discussion about the distinction between NAPs and transcription factors. To clarify it we have modified the Discussion slightly (lines 412, 417).

Reviewer #2 (Remarks to the Author):

The manuscript by Skotnicka et al describes the identification, and biochemical and biophysical characterization of the CdbA protein and how it binds DNA and the molecule di-c-GMP. Using a combination of X-ray crystallography, HDX-MS and cellular assays they defined the molecular basis for its regulation by di-c-GMP and its role in promoting cellular growth. They identified multiple mutations that fully disrupt DNA and di-c-GMP binding. They have mapped potential DNA binding sites, and showed a clear role in cellular growth.

Overall, this is an interesting manuscript, which provides unique molecular insight into the role of a novel DNA binding protein, that can be regulated by the messenger di-c-GMP. I am quite positive in my assessment, although certain aspects of the presentation could be improved to increase the clarity and accessibility to a broad audience.

Response: Thank you very much!

Major points

1. The HDX-MS experiments are well described in the methods. However how it is presented could be improved. Figure 2 that shows the HDX data and crystallographic data is very hard to interpret. First in Fig. 2b the deuterium incorporation is shown at a per residue basis, which is calculated from overlapping peptides. The data was performed in triplicate as described in methods, but no data is provided whatsoever on the error within the experiments. This needs to be incorporated into the figures/supplement.

Response: We apologize. We followed the reviewer's suggestion and changed the presentation of the HDX-MS data in the result part significantly to increase clarity (lines 177-199). We hope that the HDX data are now more readily accessible. In short, we divided the old figure 2 into two parts and now describe the crystal structure (new Fig. 2) and the HDX-MS (new Fig. 3) separately. We also changed the HDX-MS figures and have now included the HDX changes for representative peptides (Fig. 3a). As also suggested by Masson et al 2019, this is accompanied by the uptake charts of some representative peptides exhibiting perturbations upon c-di-GMP binding (Fig. 3b). Finally, in Fig. 3c,d, we mapped three representative peptides onto the dimer and tetramer structure of CdbA. We also show plots for all peptides generated by MEMHDX in the supplemental information (Supplementary Fig. 2d,e) as well as supplemental dataset containing the entire HDX dataset (Supplementary Table 2) complying with the recommendations of the HDX-MS community (Masson et al 2019).

2. Ideally the full HDX dataset should be included in excel format as suggested by the recent community guidelines as published in Nature Methods (Masson et al 2019). With these details included it would be much easier to interpret the data.

Response: We apologize for not including these data. We have now included a Supplementary Table 2 adhering to Masson et al 2019 in which we report the raw data of the HDX-MS experiments.

3. It is also not entirely clear how the structures are colored for the HDX-MS analysis of the di-c-GMP interface as all residues from 1-30 are protected, yet only a selection of those residues is mapped on the structure and described in text. The region from 12-20 is colored in Fig. 2b, yet is not described. Is this change really significant in these regions? More details on the raw deuterium incorporation and the error of the experiments would make this much clearer.

Response: We apologize for not being clear. As described in our response to comment #1 of this reviewer, we have changed the HDX-MS result part significantly to increase clarity. Also, we included all raw data in Supplementary Table S2 and Supplementary Fig. 2.

4. The overlay of the CdbA protein and its putative DNA binding interface compared to Arc is very difficult to interpret as currently presented in Fig. 2e. It would be useful to also include an electrostatic surface of CdbA to examine the feasibility of the putative DNA binding surface.

Response: Thank you very much for this suggestion. Following the suggestion of the reviewer, we have added a new panel (Fig. 2c) with the electrostatic surface of tetrameric CdbA. This analysis nicely shows that the DNA binding interface of CdbA is positively charged (line 169-170).

5. The mutants generated that disrupt DNA and di-c-GMP binding are a very nice addition to the manuscript. It is not entirely clear how different the mutants are to WT, it might be useful to compare on the same gel filtration trace the mutant vs wild type. It almost appears that the mutants are more closely mimicking the WT bound to di-c-GMP state (even though they cannot bind this molecule).

Response: We apologize for this confusion. We tried to follow the advice of the reviewer for modifying the figure. However, that did not work so well. Instead, we modified the labeling of the SEC traces to make it clearer which protein is shown in which panel. We hope that with the new panels and the table in Fig. 1d, it is clearer that the two variants (with and without addition of c-di-GMP) behave as the WT protein without c-di-GMP in the SEC experiments.

Reviewer #3 (Remarks to the Author):

In this manuscript by Skotnicka et al, the authors identified CdbAB as c-di-GMP binding proteins, solved the crystal structure of CdbA, identified residues in CdbA that were important for c-di-GMP binding, and found that mutants that did not bind c-di-GMP abolished DNA binding. Using ChIP-seq and fluorescence microscopy, the authors found that CdbA bound to the chromosome globally, and that depletion of CdbA caused defects in chromosome segregation and organization, and ultimately led to cell division defects. Because CdbA had little effect on transcription, the authors concluded that CdbA is a nucleoid associated protein, whose activity could be modulated by c-di-GMP. This is the first example that the second messenger c-di-GMP could play a role in chromosome organization and segregation. However, the authors also showed that modulating c-di-GMP levels in vivo by expressing heterologous DGC or PDE had no effect on cell growth or chromosome biology. Therefore, perhaps beyond the scope of this study, whether and how c-di-GMP modulates CdbA activity in vivo are still to be addressed.

This is a solid study involving a comprehensive set of approaches and assays. The experiments were carefully executed, the results were critically analyzed and interpreted, and the conclusions were well supported by experimental data.

Response: Thank you very much!

Minor points:

In Figure 6f, DAPI-stained nucleoid should be colocalized in the same cells to show whether the CdbA*-mCherry colocalizes with the nucleoid, to support conclusions of Line 339-341: "The two variants also failed to localize over the nucleoid in *M. xanthus* (Fig. 6f), confirming the in vitro result that they have a DNA binding defect (Fig. 6c)."

Response: Good point and thank you! We repeated this experiment and included the DAPI staining. In the new Fig. 7f (previously Fig. 6f), we have included images from these new experiments that clearly show that the mutant CdbA*-mCherry variants do not colocalize with the nucleoid while the WT CdbA-mCherry protein does.